# A cholinergic feedback circuit to regulate striatal population uncertainty and optimize reinforcement learning

Nicholas T Franklin, Michael J Frank*

Department of Cognitive, Linguistic and Psychological Sciences, Brown Institute for Brain Science, Brown University, Providence, United States

**Abstract** Convergent evidence suggests that the basal ganglia support reinforcement learning by adjusting action values according to reward prediction errors. However, adaptive behavior in stochastic environments requires the consideration of uncertainty to dynamically adjust the learning rate. We consider how cholinergic tonically active interneurons (TANs) may endow the striatum with such a mechanism in computational models spanning three Marr's levels of analysis. In the neural model, TANs modulate the excitability of spiny neurons, their population response to reinforcement, and hence the effective learning rate. Long TAN pauses facilitated robustness to spurious outcomes by increasing divergence in synaptic weights between neurons coding for alternative action values, whereas short TAN pauses facilitated stochastic behavior but increased responsiveness to change-points in outcome contingencies. A feedback control system allowed TAN pauses to be dynamically modulated by uncertainty across the spiny neuron population, allowing the system to self-tune and optimize performance across stochastic environments.

## Introduction

When tasked with taking an action in an unknown environment, there can be considerable uncertainty about which actions will lead to the best outcomes. A principled way to resolve this uncertainty is to use previous experience to guide behavior towards actions that have led to positive outcomes in the past and away from actions that have led to negative outcomes. Convergent evidence suggests that the basal ganglia can guide behavior by incorporating positive and negative feedback in a reinforcement learning process (*O'Doherty et al., 2003*; *Barnes et al., 2005*; *Frank, 2005*). However, learning can be complicated in a changing environment, as the validity of past experiences and the relationship between actions and outcomes become uncertain as well. Mathematical models suggest that it is optimal to take uncertainty into account in learning and decision making (*Yu and Dayan, 2005*; *Behrens et al., 2007*; *Mathys et al., 2011*), but it is unclear whether the basal ganglia can directly consider uncertainty in feedback-based learning.

Basal ganglia-dependent learning is often described within the normative framework of reinforcement learning (RL) following the observation that signaling from dopaminergic afferents matches the pattern of a reward prediction error (RPE) (*Montague et al., 1996*; *Bayer et al., 2007*). An RPE is the signed difference between the observed and expected outcomes and is often used in RL to generate point-estimates of action-values (*Sutton and Barto, 1998*). Phasic dopamine (DA) is thought to provide an RPE signal to striatal medium spiny neurons (MSNs) and induce learning through changes in corticostriatal plasticity (*Montague et al., 1996*; *Reynolds and Wickens, 2002*; *Calabresi et al., 2007*), with opponent learning signals in the direct and indirect pathways (*Frank, 2005*; *Collins and Frank, 2014*). Within these pathways, separate populations code for the (positive and negative) values of distinct action plans (*Samejima et al., 2005*). Multiple lines of

*For correspondence:
Michael_Frank@brown.edu

**eLife digest** One of the keys to successful learning is being able to adjust behavior on the basis of experience. In simple terms, it pays to repeat behaviors with positive outcomes and avoid those with negative outcomes. A group of brain regions known collectively as the basal ganglia supports this process by guiding behavior based on trial and error.

However, circumstances can and do change: a behavior that routinely produces a positive outcome now will not necessarily always do so in the future. The outcome of a given behavior can also vary from time to time purely by chance: even the most appropriate action can sometimes lead to a negative outcome but should be repeated again. Exactly how the basal ganglia take into account this uncertainty over behavioral outcomes to appropriately guide learning is unclear.

Franklin and Frank have now come up with a possible explanation by building on an existing computational model of the basal ganglia. Whereas the old model assumes that the rate of learning given an unexpected outcome always remains constant, in the new model learning occurs more quickly when the outcome of a behavior is uncertain. This makes intuitive sense, in that rapid learning is especially useful during the initial stages of learning a new task or following a sudden change in circumstances.

The new model proposes that a group of cells called tonically active interneurons (TANs), which release the chemical acetylcholine, enable the basal ganglia to take account of uncertainty. TANs are found in a basal ganglia structure called the striatum and have a characteristic firing pattern during important outcomes, consisting of a burst of activity followed by a pause lasting several hundred milliseconds. The model suggests that when the outcome of a behavior is uncertain, the length of this pause is increased. This boosts the activity of another group of neurons in the striatum, known as spiny neurons, and this in turn increases the rate of learning.

Franklin and Frank found that by varying the length of the TAN pause, the basal ganglia can adjust learning rates based on the degree of uncertainty over behavioral outcomes. Comparisons show that the TAN computational model optimizes the accuracy and flexibility of learning across different environments, while also accounting for findings which show that TAN lesions induce insensitivity to changes in decision outcomes. The next step is to test some of the new predictions about uncertainty experimentally.

evidence in humans and animals support this model, including optogenetic manipulations (*Tsai et al., 2009*; *Kravitz et al., 2012*), synaptic plasticity studies (*Shen et al., 2008*), functional imaging (*McClure et al., 2003*; *O'Doherty et al., 2003*), genetics and pharmacology in combination with imaging (*Pessiglione et al., 2006*; *Frank et al., 2009*; *Jocham et al., 2011*) and evidence from medication manipulations in Parkinson's patients (*Frank et al., 2004*).

Despite the substantial empirical support for RPE signals conveyed by dopamine, the simple RL mechanisms used to model the basal ganglia are inflexible in the degree to which they learn in a changing environment. RL models typically adopt a fixed learning rate, such that every RPE of similar magnitude equally drives learning. However, a more adaptive strategy in a changing environment is to adjust learning rates as a function of uncertainty, so that unexpected outcomes have greater influence when one is more uncertain of which action to take (e.g., initially before contingencies are well known, or following a change-point), but less influence once the contingencies appear stable and the task is well known (*Yu and Dayan, 2005*; *Behrens et al., 2007*; *Nassar et al., 2010*; *Mathys et al., 2011*; *Payzan-LeNestour et al., 2011*). This Bayesian perspective presents the additional challenge for basal ganglia-dependent learning: in order to take advantage of its own uncertainty over action selection, the basal ganglia would need a mechanism to translate its uncertainty into a learning rate.

Cholinergic signaling within the striatum offers a potential solution to this challenge. With few exceptions (*Tan and Bullock, 2008*; *Ashby and Crossley, 2011*), models of the basal ganglia typically do not incorporate striatal acetylcholine. Within the striatum, cholinergic interneurons are the predominant source of acetylcholine (*Woolf and Butcher, 1981*). These interneurons, also known as tonically active neurons (TANs) due to their intrinsic 2–10-Hz firing pattern, are giant, spanning large

areas of the striatum with dense axonal arborization and broad synaptic input (*Goldberg and Reynolds, 2011*). TANs appear to be necessary to learning only when flexibility is required (*Ragozzino et al., 2009*; *Bradfield et al., 2013*), suggesting that they might modulate the learning rate as a function of changes in outcome statistics (i.e., uncertainty). Similar to dopaminergic neurons, TANs show sensitivity to rewarding events and develop a learned phasic response to predictive cues (*Aosaki et al., 1994*; *Morris et al., 2004*). This response consists of a phasic burst in TAN activity followed by a pause that lasts a few hundred milliseconds (*Aosaki et al., 1995*). While the temporal pattern of the burst–pause response is temporally concomitant to the dopamine response (*Morris et al., 2004*), the unidirectional TAN response is not consistent with a bivalent RPE (*Joshua et al., 2008*) but instead is thought to provide a permissive signal for dopaminergic plasticity (*Graybiel et al., 1994*; *Morris et al., 2004*; *Cragg, 2006*).

But how would such a permissive signal be modulated by the network's own uncertainty about which action to select? Because TANs receive broad inhibitory synaptic input from local sources, including MSNs and GABAergic interneurons (*Bolam et al., 1986*; *Chuhma et al., 2011*), we hypothesized that the pause would be modulated by a global measure of uncertainty across the population of spiny neurons. Given that MSN sub-populations code for distinct action values (*Samejima et al., 2005*; *Frank, 2005*; *Lau and Glimcher, 2008*), co-activation of multiple populations can signal enhanced uncertainty over action selection, which would translate into greater inhibition onto TANs. The synchrony in the TAN response suggests a global signal (*Graybiel et al., 1994*), which can then be modulated by inhibitory MSN collaterals across a large range of spiny inputs. The TAN pause response is consistent with a signal of uncertainty that adjusts learning. First, it increases with the unpredictability of a stochastic outcome (*Apicella et al., 2009*, *2011*). Second, pharmacological blockade or lesioning excitatory input to TANs impairs learning, specifically after a change in outcome contingencies (*Ragozzino et al., 2002*; *Bradfield et al., 2013*). For an optimal learner, both increases in stochasticity and changes in outcome contingencies results in an increase in uncertainty (*Yu and Dayan, 2005*; *Nassar et al., 2010*).

Here, we augment a well-established computational model of the basal ganglia (BG) to include a mechanism by which the effective learning rate is modulated by cholinergic signaling, and where this signaling is, in turn, modulated by uncertainty in the MSN population code via reciprocal TAN–MSN interactions. In the model, cholinergic signals dynamically modulate the efficacy of reinforcement feedback, driving changes in the number of neurons available for synaptic plasticity during reinforcement, hence the effective learning rate of the network as a whole. Thus, TANs allow the basal ganglia to tailor its learning rate to its environment, balancing the tradeoff between learning flexibility and robustness to noise by adjusting learning as a function of policy uncertainty embedded in the MSN population code. We show that this behavior is consistent with a normative account using an approximately Bayesian model and that its main functionality can be simplified in algorithmic form using a modified RL model, thus spanning Marr's levels of implementation, algorithm, and computation. The model is consistent with several existing lines of evidence and makes falsifiable predictions.

## Results

We extended a previously published neural network model of the basal ganglia to examine the potential role for TANs in learning. *Figure 1* shows a graphical representation of the model, which has been adapted from *Frank, 2006* to include TANs. The basic mechanics of the BG model without TANs have been reviewed elsewhere (e.g., *Maia and Frank, 2011*; *Ratcliff and Frank, 2012*) but are summarized here before articulating the TAN mechanisms. In the network, striatonigral and striatopallidal MSNs are simulated with separate populations of 'Go' and 'NoGo' units (rate-coded point neurons), which act to facilitate or suppress selection of specific actions. These MSN units receive excitatory input from a sensory input cortical layer and a motor cortical layer representing specific candidate responses. A given response is executed when its motor cortical firing rate exceeds a threshold.

The basal ganglia participate in action selection by selectively disinhibiting thalamocortical activity for a given response while inhibiting thalamocortical activity for alternative responses, via MSN projections through different basal ganglia nuclei (*Frank, 2006*; *Humphries et al., 2006*). Activity in Go units contributes evidence toward the selection of an action in the thalamus and ultimately the cortex, while NoGo unit activity suppresses action selection and effectively provides evidence against

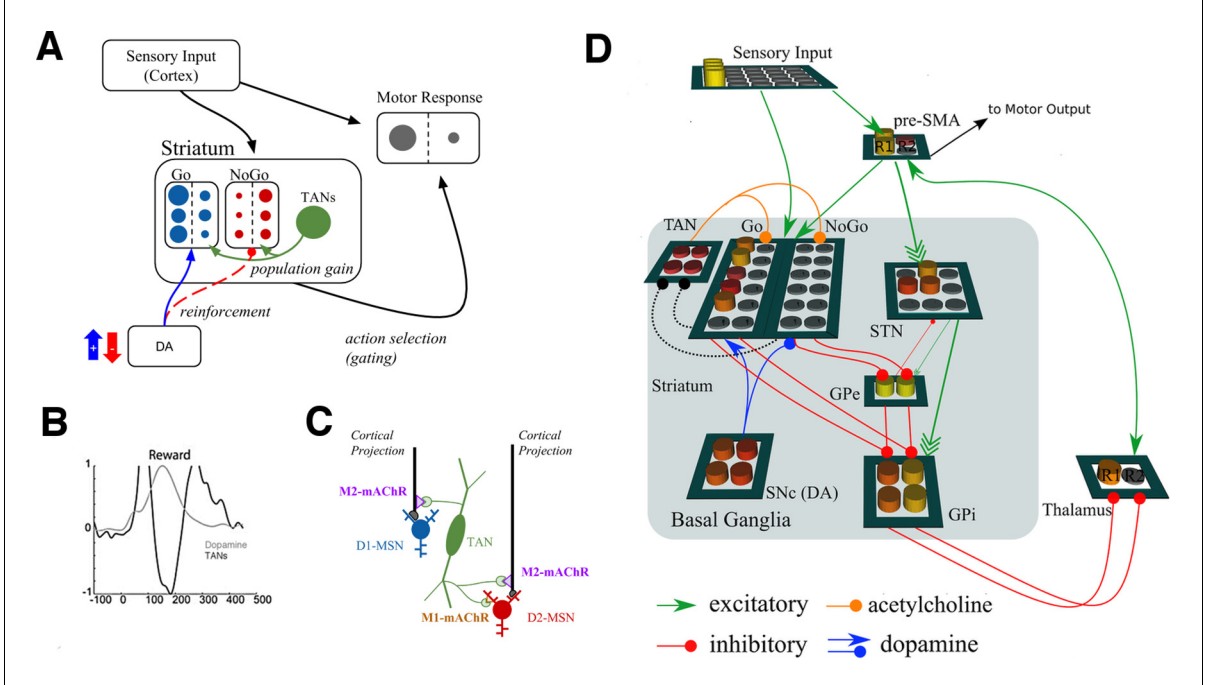

**Figure 1.** Neural network model. (**A**) A simplified diagram of the neural network. Sensory input excites candidate motor actions and corresponding Go (blue circles) and NoGo (red circles) MSN units in the striatum. Distinct columns of Go and NoGo units provide the thalamus with positive and negative evidence for alternative motor responses, learned via DA reinforcement signals. Positive prediction errors result in increased DA during feedback and an increase of Go excitability relative to NoGo units. TANs are endogenously active units that modulate MSN excitability during reinforcement feedback, altering the efficacy of the reinforcement signal. (**B**) Stereotypical TAN response is temporally concomitant to reward-related phasic DA increase (Adapted from Figure 7 part C, *Morris et al., 2004*). (**C**) Schematic representation of TAN–MSN signaling (see below). TAN firing inhibits presynaptic glutamatergic signaling of D1 and D2 MSNs through M2 receptors, but also selectively excites D2 MSNs via M1 receptors. (**D**) Sensory input provides excitatory signaling to preSMA (representing two candidate motor actions) and corresponding Go and NoGo MSN units in the striatum. Each of (here, two) motor responses is coded by distinct columns of Go and NoGo units, representing positive and negative evidence for alternative that responses, learned via reinforcement conveyed by DA signals in the SNc. The basal ganglia contribute to action selection by disinhibiting thalamocortical activity for representing thatthe response having the largest Go–NoGo activation differential. Go units inhibit globus pallidus internal segment (GPi) units, which otherwise tonically inhibit the thalamus. NoGo units have the opposite effect by inhibiting the external segment (GPe), which in turn inhibits GPi. TANs are represented as a separate layer of endogenously active units that modulate MSN excitability during the dynamic burst–pause pattern windowing the dopaminergic reward prediction error signals. This pause duration can be fixed, or sensitive to the population uncertainty of MSNs (see below). Dotted black lines correspond to proposed feedback mechanism from MSNs to TANs. The STN modulates the threshold by which actions are disinhibited by the BG, included here for consistency with prior work. DA, dopamine; MSN, medium spiny neuron; preSMA, pre-supplementary motor cortex; SNc, substantia nigra pars compacta; STN, subthalamic nucleus.

particular responses. The difference in Go and NoGo unit activity for each candidate action effectively provides a decision variable, each represented in distinct columns of cortico-BG-thalamic circuits. The greater the relative Go–NoGo activity difference for a particular response, the more likely the corresponding column of thalamus will be disinhibited, and hence the corresponding action selected. Initially, network choices are determined by random synaptic weights favoring one or the other action and the additional intra-trial noise in unit activity favoring exploration of alternative actions, but choices stabilize via dopamine-dependent learning, as described next.

Dopamine modulates excitability in the Go and NoGo pathways and has opposing effects on the two populations. D1 receptor activity simulated in the Go units results in increased excitability while D2 receptor activity simulated in NoGo units has the opposite effect. Consequently, an increase in dopamine leads to increased Go activity relative to NoGo activity; this can influence both the choice process through acute changes in excitability and the learning process through consequent changes in synaptic efficacy during phasic changes in dopamine-related to positive and negative outcomes.

Several predictions of this model have been validated empirically across species, including effects of pharmacological manipulations, genetic variants, and optogenetic manipulations (**Collins and**

*Frank, 2014* for review and quantitative simulations) but as of yet, it has not explored the role of TANs. Furthermore, while it can account for general effects of dopaminergic manipulation on probabilistic reversal learning (*Frank, 2005*), it fails to appropriately modulate its learning as a function of environmental noise and internal action uncertainty. Finally, while BG RL models depend on DA-driven learning, they do not provide a mechanism that would allow striatal learning to occur selectively during reward prediction errors as opposed to various other factors that could also change DA levels (e.g., *Howe et al., 2013*). We investigated whether the TAN pause, by windowing dopaminergic RPEs, can provide such a permissive signal on learning.

The stereotypical TAN burst–pause response modulates excitability and presynaptic corticostriatal signaling in both MSN populations (*Gerfen and Surmeier, 2011*). In our model, a TAN pause was simulated at the onset of feedback to create a window around the dopaminergic reinforcement signal (*Morris et al., 2004*); to isolate the effects solely attributable to learning, TAN activity levels were constant during action selection itself (*Figure 2*, left; see Materials and methods). Cholinergic

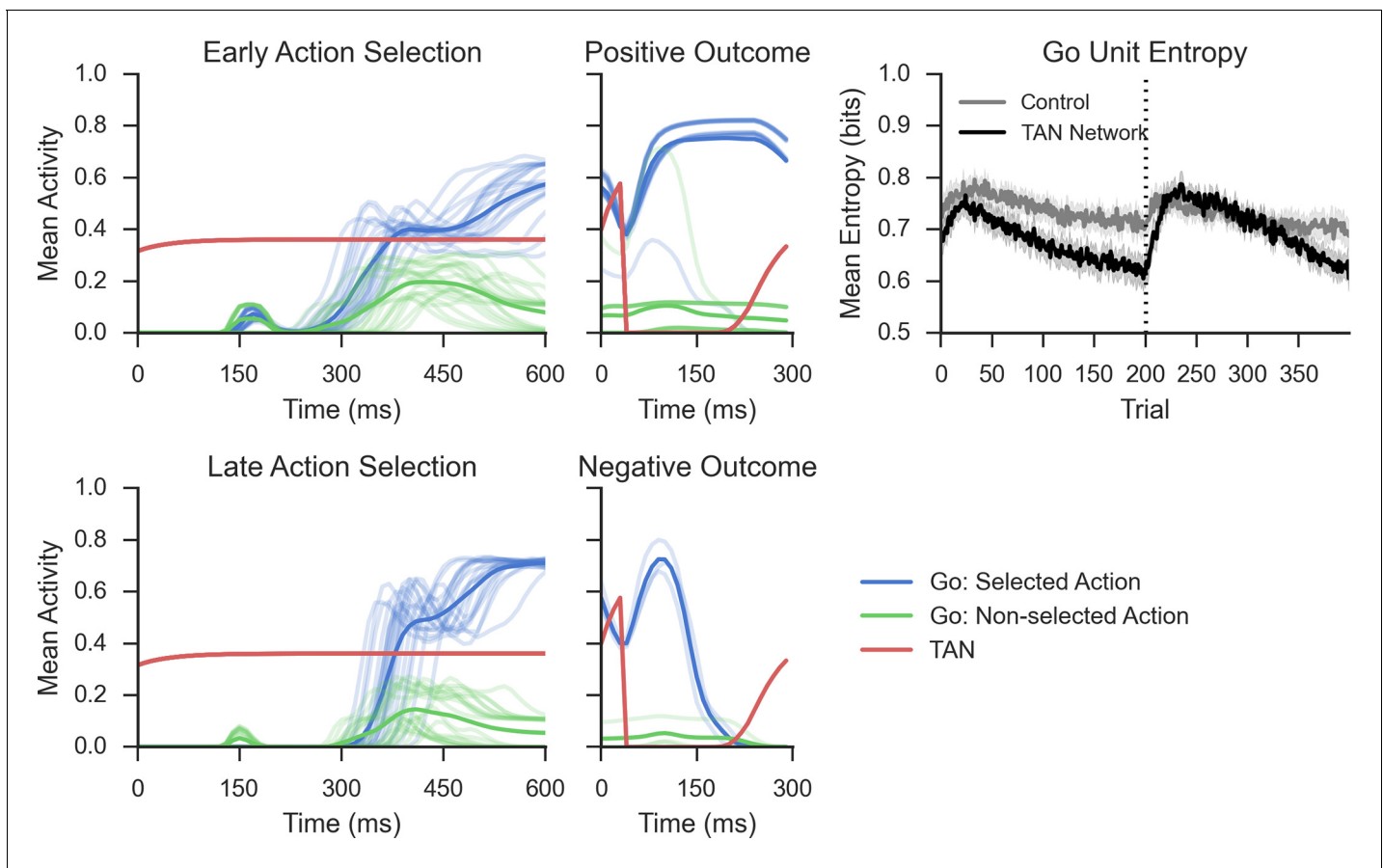

**Figure 2.** Network activity in a probabilistic reversal learning task. *Top left*: Mean within-trial normalized firing rate across population of Go units (simulated striatonigral MSNs) and TANs during action selection during the first epoch of training ('early'). Individual traces represent the mean population activity in a single trial and are collapsed across response for Go units. In the examples shown, Go unit activity ultimately facilitates the selected response (due to disinhibition of thalamus; not shown, see *Frank, 2006*). *Bottom left*: Mean within-trial firing rate during action selection in the last epoch of training ('late') prior to reversal. As a consequence of training, an increased differential in Go activity between the selected and non-selected response results in low action-selection uncertainty. *Center*: Mean firing rate for Go and TAN units during reinforcement for both correct (*top*) and incorrect (*bottom*) trials. TAN pauses occur for both types of feedback, providing a window during which the dopaminergic signals differentially modulate Go unit activity and plasticity (long-term potentiation vs long-term depression). *Top right*: Entropy in population of Go units across training trials. Population-level entropy declined over time prior to reversal (trial 200; dotted line) as the stochastic population of simulated neurons learned the correct action. Following reversal, entropy rose briefly as the network starts to activate the alternative Go response while the learned one still remained active for some time, after which entropy declines once more. This dynamic modulation of entropy is more pronounced in a network with simulated TANs than a control network without TANs. MSNs, medium spiny neurons; TANs, tonically active neurons.

signaling within the striatum is extensive and complex, involving a diversity of cholinergic receptors. For tractability, we mainly focused our investigation on the effects of TAN signaling on MSN excitability, simulating the effects of M1- and M2-muscarinic acetylcholine receptors on MSNs and nicotinic receptors on GABAergic interneurons (we further explore the ACh effects on DA release in the final section). Changes in muscarinic activity modulate MSN excitability during TAN pauses, allowing the MSN population to be more or less excitable during reinforcement (see Materials and methods for details of biophysics and implementation in our model). Longer (or deeper) TAN pauses increase corticostriatal input and hence MSN activity in response to reinforcement, compared to shorter TAN pauses.

We conducted a series of simulations in which this neural network was confronted with a two alternative, forced choice reversal learning task. On each trial, one of two input stimuli was presented and the network had to select one of two responses. For the initial simulations, one response was rewarded 80% of the trials and the other 20% of the trials. (Subsequent simulations described later varied the stochasticity of the environment.) The network was trained on 20 epochs of 20 trials for each stimulus, or 400 trials total. Performance was probed prior to the first training epoch as a baseline and following the completion of each training epoch. The first 10 training epochs comprised the 'acquisition' phase in which the reward schedule remained constant. Adaptive behavior in this context demands learning from unexpected outcomes during initial stages of the task but ignoring spurious outcomes (i.e. the 20% of negative outcomes) once the contingencies are well known. The reward contingencies were then reversed for the remaining 10 epochs ('reversal' phase) such that the response with the lower value during acquisition was now rewarded in 80% of trials. An adaptive learner would show renewed sensitivity to unexpected outcomes when the contingencies change, and then stabilize again asymptotically.

Because the TAN pause results in MSN disinhibition, it modulates their excitability during outcome presentation (when phasic dopaminergic signals are presented) via modulation of M1 and M2 receptors (see Materials and methods). TAN pauses elicit greater activation of the most excitable units, which then, due to activity-dependent plasticity, are most subject to dopaminergic learning. In turn, the resultant synaptic weight changes affect the subsequent distribution of active MSNs during action selection. This distribution can be quantified by the uncertainty in action selection across the MSN population, which we define here as the Shannon's entropy based on the firing rates y of units associated with each action a across all time points t during action selection within a trial:

$$H = -\sum_t \sum_a p_a(t) \log_2 p_a(t),$$

where $p_a(t)$ reflects the population's probability assigned to selecting action a based on normalized firings rates coding for that action (see Materials and methods). A population with low entropy has a few strongly active units representing primarily the dominant motor response, whereas a population with high entropy has many weakly active units representing multiple alternative responses. As the network learns, entropy across Go units declines over time and the network becomes more likely to strongly activate Go units associated with the correct response, and less likely to activate those for the opposing response (*Figure 2*, top right). Following the reversal, the entropy of the Go units increases as the network un-learns the previous correct response (making those units more weakly active) while beginning to activate those associated with the alternative response. While this pattern is evident in networks without TANs, it is more pronounced in TAN networks: the selective modulation of excitability during outcomes amplifies the effect of dopamine on Go activity for the dominant response, progressively differentiating from the less rewarded response and leading to low entropy Go representations (*Figure 2*, top right).

The decline in MSN entropy translates into reduced stochasticity in action selection, as the MSN population becomes dominated by a single strongly active population, providing stronger evidence for the frequently rewarded action. Consequently, TAN networks exhibit higher accuracy when a stochastic task is well learned than a network without TANs (*Figure 3*, left). However, this property alone reduces the flexibility of the network to respond to change points, because a single strongly active population is harder to unlearn, and the network will perseverate in selecting the previously learned response. This tradeoff will become evident below, where we show that TAN pause durations can be dynamically modulated to optimize the tradeoff between flexibility and stability.

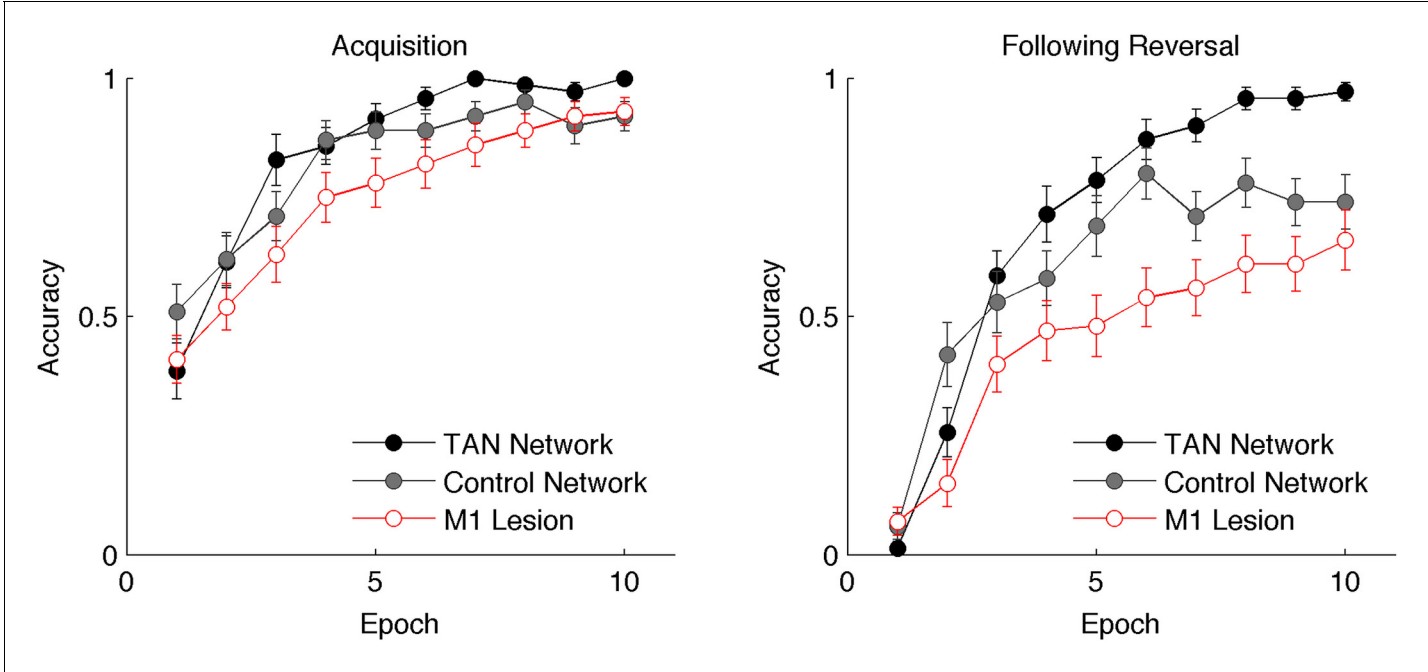

**Figure 3.** Performance of the neural network on reversal learning task. *Left*: Accuracy during acquisition of probabilistic contingencies. During initial acquisition, accuracy for networks simulated with TANs (black) is similar to accuracy in control networks that do not contain TANs (grey). Simulated M1 blockade (red) does not meaningfully impair performance during acquisition. *Right*: Accuracy following reversal. Networks with TANs (black) reach higher asymptotic performance than control networks (grey). Networks with simulated M1 blockade (red) show pronounced performance impairments. TANs, tonically active neurons.

To directly study the impact of TANs on learning, we first parametrically varied the duration of the TAN pause exogenously (below we explore how this pause may be endogenously self-regulated by the network). Varying pause duration across networks reveals a behavioral trade-off between learning speed following probabilistic reversal and asymptotic accuracy (*Figure 4*, top row). During acquisition, learning speed is similar across a range of TAN pause durations, but networks with long pauses exhibit higher levels of asymptotic accuracy. Long pauses facilitate a disproportionate increase in synaptic efficacy for the most frequently rewarded response despite the occasional spurious outcome (*Figure 5*, bottom right), driving MSNs to focus their activity to a single dominant action (as quantified by MSN entropy; *Figure 4*, bottom). However, this same property makes such networks slow to reverse. Conversely, networks with short TAN pause durations learn more quickly following reversal but at a cost of asymptotic accuracy. Mechanistically, this occurs as short TAN pauses elicit weaker activation of spiny units, and hence elicit less activity-dependent plasticity, leading to a smaller learned divergence in synaptic weights between alternative responses (*Figure 5*, bottom right). This stunted divergence allows for continued co-activation of multiple response alternatives during subsequent action selection, and hence more stochastic choice, but this same property supports the network's ability to quickly reinforce alternative actions following change points. Thus, although in these simulations the TAN pause modulates excitability only during window of the phasic dopaminergic reinforcement signal, it influences which MSNs learn and hence the population activity and action selection entropy thereof during subsequent choices.

The importance of this trade-off is most evident across different levels of predictable stochasticity, which was manipulated by varying the reward schedule. In a probabilistic reversal learning task in which the optimal action was rewarded with an 85% reward rate (suboptimal action rewarded 15%), networks with a short TAN pause outperformed networks with a longer TAN pause (*Figure 6*, left). Accuracy (percent choice of the optimal action) across all trials was 78% (SD=7%) for the shortest pause compared with 70% (SD=5%) for the longest TAN pause [$t(148) = 7.3$, $p < 3 \times 10^{-10}$, Cohen's d = 1.2]. In a more stochastic task with only 40% reward rates for the optimal action (10% for the suboptimal action), the pattern is reversed: long pause networks obtained an overall accuracy level of

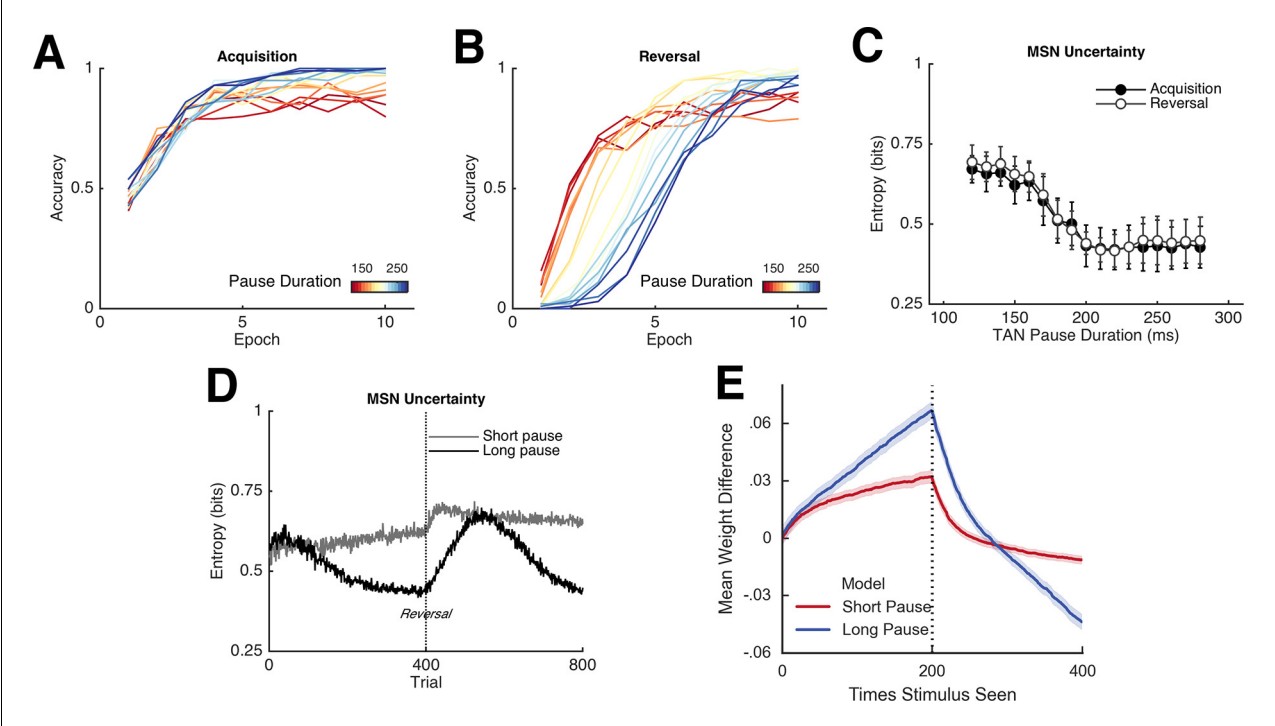

**Figure 4.** Network behavior as a function of TAN pause duration. (A, B): Accuracy of the neural network simulations over a range of fixed pause durations (120–280 ms) during initial acquisition (A) and following reversal (B). Networks with short TAN pauses (red) learned more quickly following reversal than networks with a long TAN pauses (blue) but did not achieve the same level of asymptotic accuracy during either acquisition or following reversal. (C) Final MSN entropy after training for both acquisition and reversal, as a function of TAN pause duration. Longer pauses elicited lower entropic MSN representations, facilitating asymptotic performance, whereas the higher entropy in short pause networks facilitated faster reversal. (D) Action selection uncertainty across MSN population across all trials for both stimuli. Entropy in networks with a long TAN pause (black) declines over time and reaches asymptotic level both prior to and following reversal. (E) Mean difference in corticostriatal Go weights (synaptic efficacy) coding for the more rewarded versus less rewarded response (defined during acquisition) are shown across training trials for a single stimulus, in an 85:15% reward environment. Greater accumulation in weight differences result in more deterministic choice but difficulty in unlearning. Data shown for networks trained on with a long (270 ms) or short (150 ms) fixed duration TAN pause. Reversal at trial 200 denoted with dotted line. MSN, medium spiny neurons; TANs, tonically active neurons.

53% (SD = 10%) compared to 65% (SD = 10%) for long pause networks [t(148) = 8.1, p<3x10$^{-13}$, Cohen's d = 1.3] (*Figure 6*, center). In both environments, overall accuracy varies parametrically with pause duration and degrades gracefully. These simulations show that, for each environment, the optimal pause duration is dependent on the cost of sensitivity to noise and the need for flexible behavior. The high MSN entropy generated by a short TAN pause allows for increased flexibility, which is advantageous in a highly deterministic environment with a single reversal, where spurious negative outcomes are less common, lowering the cost of sensitivity to noise. In contrast, the lower entropy MSN representation generated by longer TAN pauses (due to divergent weights) is advantageous when the task is highly stochastic. The ability to ignore spurious negative outcomes allows the network to maintain a stable estimate of an action value, which can be more important than behavioral flexibility. As a result, the performance of a network with fixed pause duration is dependent on the environmental statistics.

## Adaptive tuning of TAN pauses and learning rates using MSN population entropy

A principled way to balance this trade-off is to use the learner's uncertainty over its action values to anneal the learning rate, similar to the gain in a Kalman filter (*Yu and Dayan, 2005*). For an ideal observer, the proper measure of this uncertainty depends on the parameterization of the generative processes of the task, which may require several levels of hierarchical representations

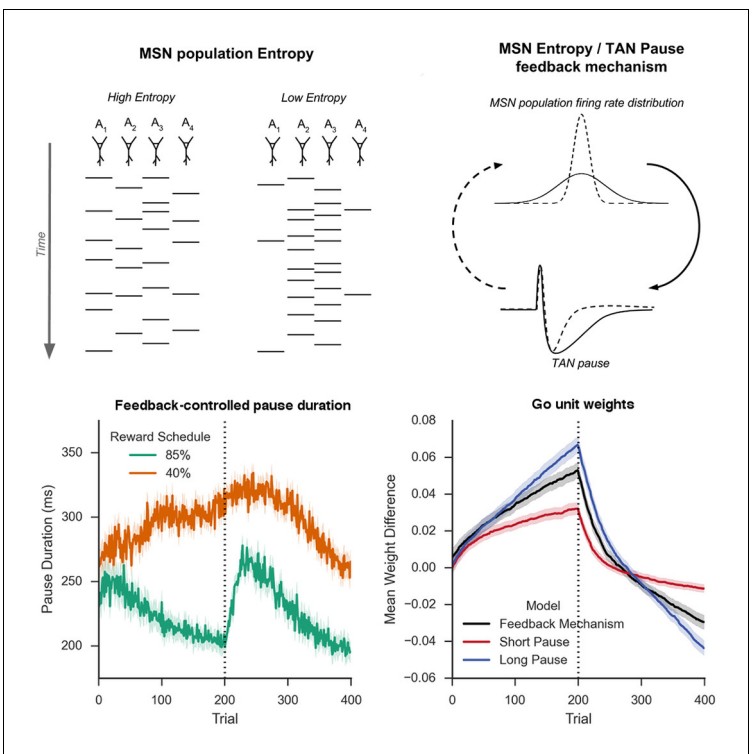

**Figure 5.** Schematic Representation of MSN entropy and self-tuning mechanism. *Top left*: Action potentials (horizontal lines) for a population of neurons with both high and low population entropy are shown across time. In the example, both entropy conditions produce the same number of action potentials in the displayed time window. In the population with high entropy, action potentials are distributed equally across the neurons thus reflecting more competition across MSNs representing different behavioral actions, and hence uncertainty about which action to select. In the population with low entropy, spike rates are concentrated in fewer neurons. *Top right*: Homeostatic feedback mechanism. The population entropy across spiny neurons can influence the duration of TAN pauses, which reciprocally influences the excitability of spiny neurons and hence entropy. This feedback control system optimizes the TAN pause duration and allows the population to be dynamically sensitive to unexpected outcomes. A high-entropy MSN population firing rate distribution (top, black line) leads to a longer TAN pause (bottom, black line), increasing weight divergence and creating downward pressure on MSN entropy. Low MSN entropy (top, dotted line) decreases TAN pause duration (bottom, dotted line) creating upward pressure on MSN entropy. *Bottom left:* Feedback-mechanism self-regulates TAN pause duration (ms) as a function of entropy across learning. Pauses become shorter for more deterministic environments as a function of learning, preventing divergence in synaptic weights and over-learning, but then increase at change points. Reversal at trial 200 denoted with dotted line. *Bottom right:* Feedback mechanism adaptively regulates synaptic weights to facilitate learning. Mean difference in corticostriatal Go weights (synaptic efficacy) coding for the more rewarded versus less rewarded response (defined during acquisition) are shown across training trials, in an 85:15% reward environment. Greater accumulation in weight differences result in more deterministic choice but difficulty in unlearning. Data shown for networks trained on with a long (270 ms) or short (150 ms) fixed duration TAN pause, as well as a network employing an entropy-driven feedback mechanism. Reversal at trial 200 denoted with dotted line. MSNs, medium spiny neurons; TANs, tonically active neurons.

(*Behrens et al., 2007*; *Mathys et al., 2011*) that may be unavailable in a local striatal network. However, as noted above, entropy over multiple MSN populations can be used as a proxy as it can be interpreted as the uncertainty with which the network selects an action. Although we have thus far described this entropy as an observable variable to provide an explanation for how TAN pauses affect the trade-off between flexibility and stability because it can be read out from the distribution of MSN firing rates, we hypothesized that this same measure may be used to reciprocally influence TAN pauses and hence learning. In this way, the system could self-detect its own uncertainty in a way that (unlike any given fixed pause duration) is not dependent on the parameterization of a particular task and is implicitly available in the population response.

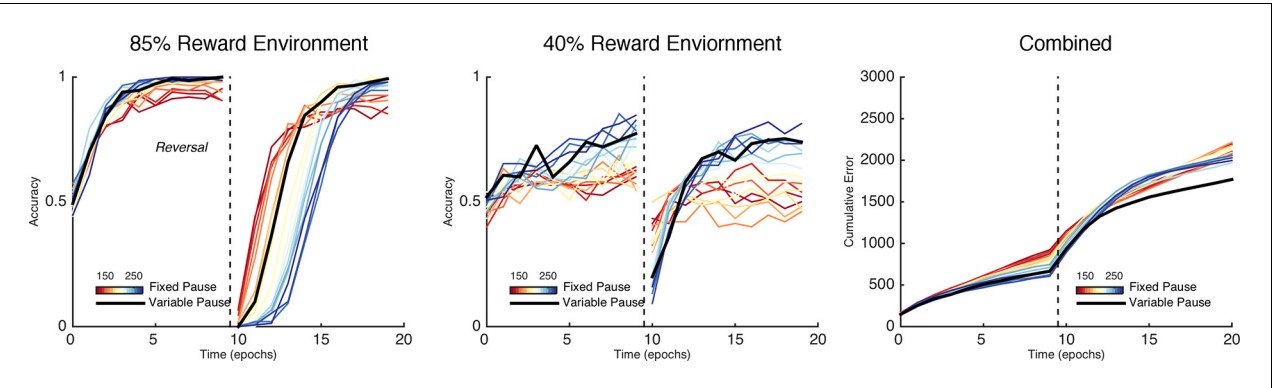

**Figure 6.** Learning in the neural network across multiple reward schedules. *Left*: Networks with fixed short TAN pauses across training (red) exhibited higher performance in an environment with 85:15% reward contingencies for the optimal and suboptimal actions, compared with networks with long TAN pauses (blue), due to increased learning speed following reversal. *Center*: Networks with long TAN pauses (blue) were able to acquire a task with 40:10% contingency, unlike networks with a short TAN pause (red). Networks with variable pause durations as a function of MSN entropy are shown in black, with good performance in both environments. *Right*: Across both reward schedules, networks in which TAN pause duration was allowed to vary with MSN entropy (black) made fewer errors than networks of any fixed duration. Reversal is denoted with dotted line in all panels. MSNs, medium spiny neurons; TANs, tonically active neurons.

Given their size and broad connectivity across large areas of striatum spanning many MSNs, it is plausible that TANS are sensitive to population MSN entropy or a correlate thereof by summing activity over multiple neurons coding for different actions. As noted earlier, TANs receive substantial GABAergic innervation from medium spiny neurons (*Bolam et al., 1986*; *Chuhma et al., 2011*; *Gonzales et al., 2013*). While the initiation of a TAN pause is dependent on thalamic and dopaminergic signaling (*Aosaki et al., 1994*; *Ding et al., 2010*), we propose that local signaling from MSNs can modulate the response by elongating the duration of the pause (or simply further reducing the firing rate during the pause) through direct inhibition. If TANs are sensitive to MSN entropy and can modulate their pause durations accordingly, they may provide a mechanism to adjust to changes in uncertainty.

To assess this potentially adaptive mechanism, we first conducted simulations in which the TAN pause was dynamically modulated by an analytical computation of MSN entropy (see Materials and methods); below, we will consider a more mechanistic implementation by which TANs respond to local circuits that approximate entropy. These simulations confirmed that dynamic modulations of TAN pause as a function of MSN mitigated the trade-off between flexible learning and sensitivity to noise. When trained on an 85% reward schedule, the network with a variable duration pause learned relatively quickly following reversal while still achieving a high level of asymptotic accuracy (*Figure 6*, left). When trained on a 40% reward schedule, performance was comparable to the best network trained with a long fixed duration pause (*Figure 6*, center). While in both the cases it was possible to find a particular fixed duration pause that can perform as well as a variable pause duration, networks with entropy-modulated pause durations performed better across both reward schedules than any fixed pause network [Accuracy $\sim N(\overline{\mu}, |\overline{\sigma})$; $E[\| \mu_{\text{variable}}\|_1 - \max \| \mu_{\text{fixed}}\|_1] = 5.76\%$; $p(\|\overline{\mu}_{\text{variable}}\|_1 \leq \max \|\overline{\mu}_{\text{fixed}}\|_1) = 1.03\text{x}10^{-3}$]. Thus, while a fixed TAN pause may perform well in any one environment, any given fixed setting is suboptimal when the environment is unknown. Varying TAN behavior with MSN entropy allows the network to learn robustly over a wider range of environment statistics, thereby allowing learning rates to be sensitive to the network's own uncertainty.

This behavioral flexibility occurs as the feedback mechanism induces longer pauses during periods of higher uncertainty. In the 85% reward environment, the feedback mechanism results in dynamic behavior: TAN pauses last greater than 250 ms both at the beginning of the task and following reversal, while pauses reach 200 ms immediately prior to reversal and at the end of the training session when less learning is required (*Figure 5*, bottom left). As such, the long TAN pause early in acquisition results in increased synaptic efficacy for Go units associated with the dominant response relative to the suboptimal response (*Figure 5*, bottom right). As the entropy and the pause

duration decline, the rate of change in synaptic efficacy slows, resulting in faster reversal learning as the network can unlearn the previous reward association more quickly. In contrast, uncertainty remains elevated in the in the highly stochastic 40% reward environment and the TAN pause consequently remains elevated throughout training (*Figure 5*, bottom left). As such, the feedback mechanism results in changes in synaptic efficacy that are similar to changes induced by a long fixed duration TAN pause.

## Normative and algorithmic descriptions of TAN pause behavior

Before further analyzing biophysical mechanisms within the model—including a mechanism for computation of entropy, the effects of M1 receptor manipulations, and additional roles of the post–pause burst on DA release—we first develop higher level (computational and algorithmic) models that summarize the key trade-off identified above in functional terms (*Marr and Poggio, 1976*). While the neural network model makes empirical predictions at the biophysical level, the core computational and algorithmic problems solved by the network can be imbedded in simpler formulations. To describe the computational and algorithmic problems solved by the network, we compared the network's behavior with two models: (1) an approximately Bayesian model that considers the higher-level computational problem of learning under uncertainty, and (2) an algorithmic model that more closely matches learning in the basal ganglia mechanistically, modified from *Collins and Frank (2014)* to include the role of TANs.

The core computational problem solved by the addition of TANs in the network addresses how to integrate noisy experiences in a changing environment. *Behrens et al. (2007)* noted that in a changing environment, a hierarchical representation of volatility can be used to adjust the learning rate in an optimal way. If rewards are distributed probabilistically with rate $r$ and changes from trial to trial, an optimal agent can estimate the volatility of $r$ as well as their distrust in the trial-to-trial volatility of the reward rate. It is unlikely that this hierarchical inference is implemented in the basal ganglia; thus, we consider here an approximation by which this computation need not be performed explicitly but where the striatum has access to its own uncertainty and adjust its learning rate accordingly.

In the tasks simulated, rewards were of equal value and delivered stochastically. In a stationary task with binomial outcomes, the posterior distribution of the reward rate for an optimal learner is a Beta distribution (*Daw et al., 2005*) parameterized by the counts of rewarded and non-rewarded trials. Because reversal tasks are, by definition, non-stationary, the Beta distribution does not represent the exact posterior distribution of expected values and is slow to adjust to a reversal: it becomes too certain about reward contingencies. A Bayesian treatment allowing for the possibility that the outcome statistics can change can be approximated using a mixture distribution combining the posterior of expected values with a uniform distribution (*Nassar et al., 2010*). In this approximation, the mixture component is the probability a change has occurred to some other unknown contingencies. Instead, we use a multiplicative parameter $\gamma$ to decay the counts of the Beta hyper-parameters towards the prior after each trial, an approximation that has been used previously to model rodent and human behavior (*Daw et al., 2005*; *Doll et al., 2009*). Decaying the counts multiplicatively maintains the mode expected value for each action but increases the variance (uncertainty) of the distribution. This effectively reduces the model's confidence of the expected value without changing its best estimate. Hence, a faster decay rate allows for greater effective learning rate, analogous to the mechanism in the BG model, whereby MSN weights are prevented from overlearning.

We simulated the Bayesian model with both fixed values of $\gamma$ as well as a $\gamma$ that varied as a function of entropy in action selection. Models with fixed values of $\gamma$ are analogous to neural networks with a fixed duration TAN pause, both constituting static strategies to balance asymptotic accuracy and learning speed. Varying $\gamma$ as a function of action selection entropy is similar to the strategy employed by the neural network, as MSN entropy corresponds to the decision uncertainty of the network.

Over both manipulations, the Bayesian model shows the same qualitative pattern of behavior as the neural network (*Figure 7*, right). For fixed values of $\gamma$, we found the same tradeoff between asymptotic accuracy and learning speed following reversal. Likewise, varying the decay rate with uncertainty mitigated the effects of the trade-off, facilitating faster learning following reversal than possible with a slow, fixed decay rate without the cost of asymptotic accuracy associated with high, fixed decay (*Figure 7*, right, black line). Given that it is an idealized statistical model and not

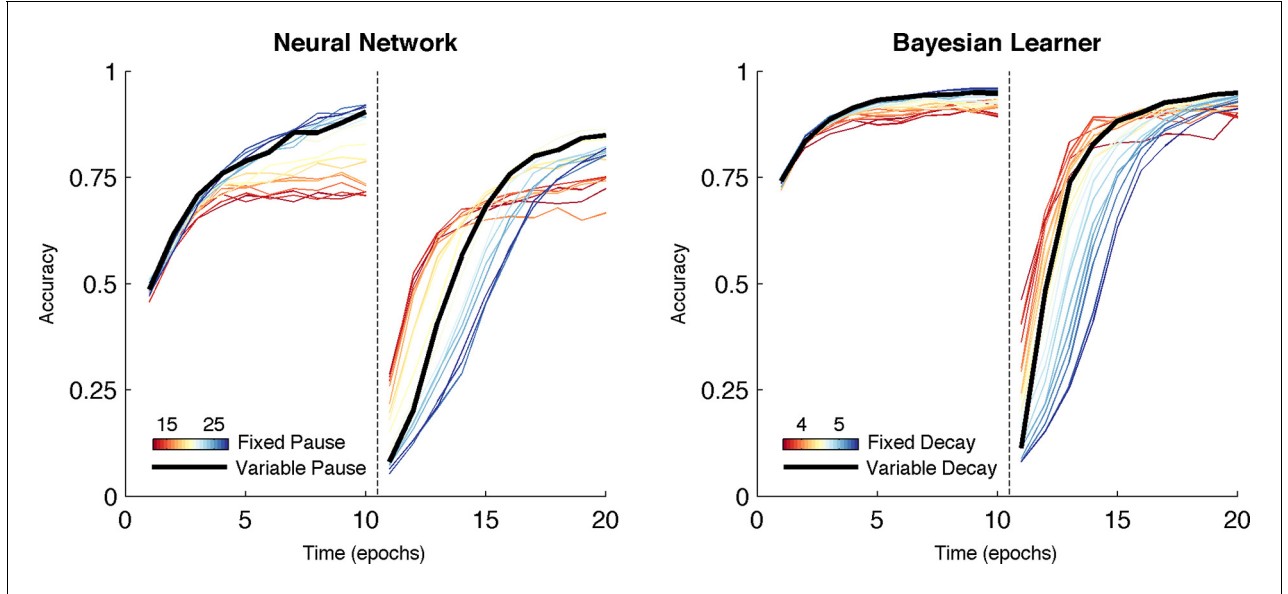

**Figure 7.** Comparison of behavior between neural network and Bayesian learner collapsed across multiple reward schedules. The performance of the Bayesian learner (i) is qualitatively similar to the performance of the neural network model (*left*). A slow decay rate in the Bayesian learner (*right, blue*) has the same effect as a long TAN pause (*left, blue*) and results in higher asymptotic accuracy at a cost of slower learning following reversal. A fast decay rate (*right, red*) has the same effect as a short TAN pause (*left, red*) and results in fasters learning following reversal with lower asymptotic accuracy. Varying the decay rate and pause duration with entropy in the Bayesian learner and neural network, respectively, mitigates the trade-off. Reversal is denoted with dotted line in both panels.

implementational, the Bayesian learner responds with an overall higher level of accuracy than the neural network. Unlike the Bayesian learner, the neural network does not learn an estimate of each action value directly but instead learns relative preferences for actions. Nevertheless, overall, the Bayesian learner provides a normative description of the neural network as the effect of manipulating the decay parameter $\gamma$ is qualitatively similar to what we see when manipulating TAN duration, suggesting that TAN duration affects the uncertainty of representations in MSNs.

The OpAL model proposed by *Collins and Frank (2014)* provides a more algorithmic summary of the basal ganglia network, as an expansion of an actor–critic model to include opponent (D1/D2) actor values. The OpAL model provides a normative advantage over traditional RL while quantitatively capturing a variety of data across species implicating opponent processes in both learning and action selection, where dopamine manipulations affect the asymmetry with which humans and animals make decisions, in a model with few free parameters. In the OpAL model, reward prediction errors are computed within a critic that evaluates the expected value of each state, and are used to update these values and train the G and N actors, summarizing the population activity of Go and NoGo units with point values (see Materials and methods). To simulate the effect of TANs in OpAL, the G and N weights were decayed multiplicatively after each trial by either a constant rate (to simulate the effects of a fixed-duration TAN pause), or as a function of G and N entropy (to simulate the effects of the proposed feedback mechanism). A fast rate of decay (closer to zero) simulated a short TAN pause whereas a slow rate of decay (closer to one) simulated a long TAN pause.

Simulations over a range of fixed rates of decay shows a similar pattern as in the neural network and Bayesian learner, as slow rates of decay showed high asymptotic accuracy but with a decreased learning speed following reversal in a deterministic environment (*Figure 8*, left). Likewise, a faster rate of decay showed a similar pattern to networks with short TAN pause durations and showed increased flexibility after reversal at a cost of asymptotic accuracy. As in the neural network, the trade-off between asymptotic accuracy and speeded learning after reversal was marked by a divergence in the G weights (*Figure 8*, right). For models with a slow decay, the difference between G weights for the initially rewarded response and the initially sub-optimal response was much more

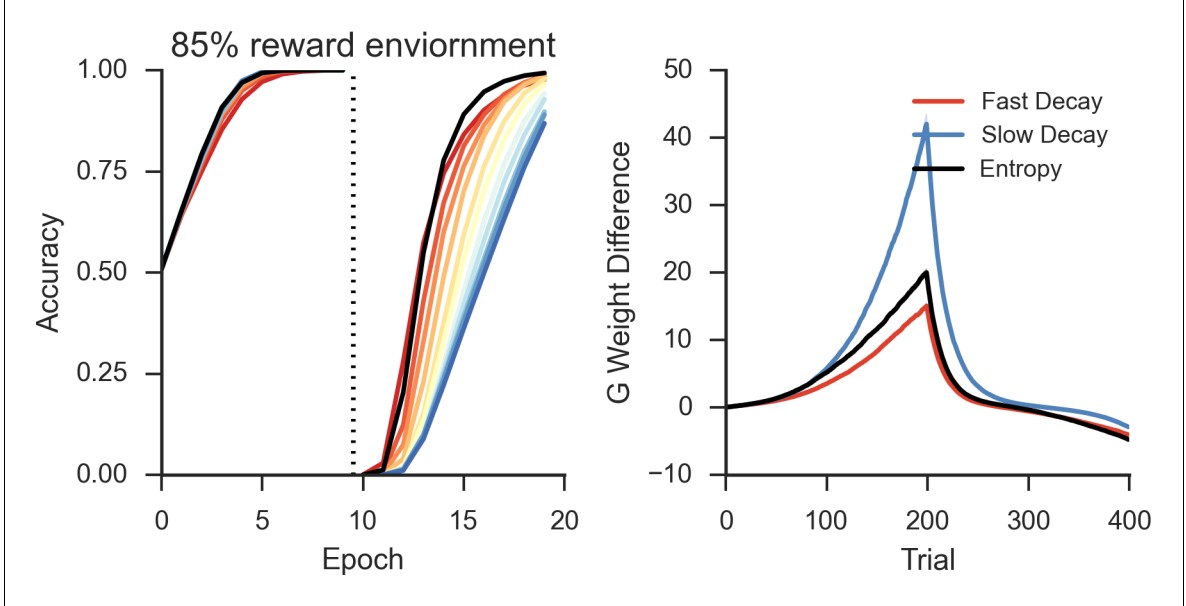

**Figure 8.** Peformance of OpAL in an 85% reward environment. *Left:* Fast decay in synaptic weights (red lines) result in lower asymptotic accuracy but speeded learning following reversal while slow weight decay (blue lines) result in the opposite pattern. A model that varies the decay rate with policy uncertainty (black line) mitigates this trade-off. *Right*: Slow decay rates result in a large divergence between G weights for the two possible actions prior to reversal (trial 200) that correlates with high asymptotic accuracy and slower learning speed following reversal, as this divergence must be unlearned. An entropy modulated decay rate shows a high initial rate of divergence sufficient to improve accuracy but slows as the model learns the task, facilitating reversal.

pronounced than models with a faster decay. This divergence declined more rapidly in models with a slow decay, allowing these models to unlearn the previously rewarded response more quickly.

Allowing the decay rate to vary as a function of uncertainty also showed the same pattern of behavior found in the neural network, mitigating the trade-off and resulting in both high asymptotic accuracy as well as speeded learning following reversal (*Figure 8*, left). Overall, varying decay with entropy resulted in higher reward rate compared with the next best performing model with fixed decay [t(198)=3.7, p<3x10$^{-4}$, Cohen's d = 0.5]. Varying decay with entropy also optimized the divergence of $G$ weights relative to fixed decay. The initial rate of divergence of the variable model is similar to that of a slow decay model, improving asymptotic accuracy. This rate of divergence declines as the model learns, preventing overlearning and facilitating faster learning following reversal. This is the same pattern observed in the divergence of Go weights in the neural model (*Figure 5*, bottom right), where varying pause duration with entropy optimized the divergence of weights relative to fixed duration pauses.

## A local mechanism for entropy modulation of TAN duration

In the neural network simulations we proposed that TANs may have access to entropy over MSNs, given that they span large regions of the striatum and receive inputs from many MSNs and GABAergic interneurons. However, the biophysical details by which spiny neuron modulate TAN activity are not fully understood. Direct transmission of entropy may not be trivial as population entropy is a nonlinear function of the units' activity. Here we consider a more explicit mechanism by which TANs directly approximate MSN entropy through synaptic integration. First, consider the minimalist case of just two MSNs coding for alternative actions. Here, the entropy is high when both are active or inactive, but low when either of the two units alone is active. In terms of a Boolean function, the entropy of the two MSNs is the logical opposite of an 'exclusive OR', a non-linear problem that typically requires interneurons to detect (*Rumelhart and McClelland, 1986*). While one potential mechanism for TANs to detect MSN entropy is by including interneurons, this non-linear detection is linearly solvable if the problem is split into two separate detections: the detection of when both neurons are active and the detection of when neither is active (*Figure 8*, top left). Consequently, a

potential approximation for MSN entropy would be the detection of coincident activity (or inactivity) of pairs of MSNs. While an active readout of joint inactivity is implausible, this problem is solved by the opponent nature of MSN pathways in the BG (*Frank, 2005*; *Kravitz et al., 2012*), where the TANs could detect either coincident activity of two D1 MSNs signaling Go activity or two D2 MSNs signaling NoGo activity.

This scheme would require a synaptic organization such that two MSNs associated with different motor responses synapse close together on the dendrite of a TAN to facilitate coincidence detection (*Figure 9*, top right). Coincidence detection between distinct signals is thought to be an important mechanism both for cellular plasticity (*Wang et al., 2000*) as well as the coordination of sensory input (*Kapfer et al., 2002*). If synaptic signaling from both MSNs is needed to propagate an action potential to the cell body, then the pair of synapses can be thought of as a coincidence detector. In logical terms, the synapse pairs perform Boolean 'AND' detection (*Figure 9*, top left). Several pair of D1 synapses located on the dendrites of a single TAN could result in the summation of these signals in the cell body of the TAN, approximating half of the entropy function signaling that there is evidence for multiple motor responses. Similarly, multiple pairs of D2 synapses contributes to the other half of the entropy function, signaling that there is evidence against multiple responses. Thus, these two measures could effectively approximate decision uncertainty in the MSN population and are detectable by a single TAN via direct synapses from MSNs. In principle, other mechanisms within

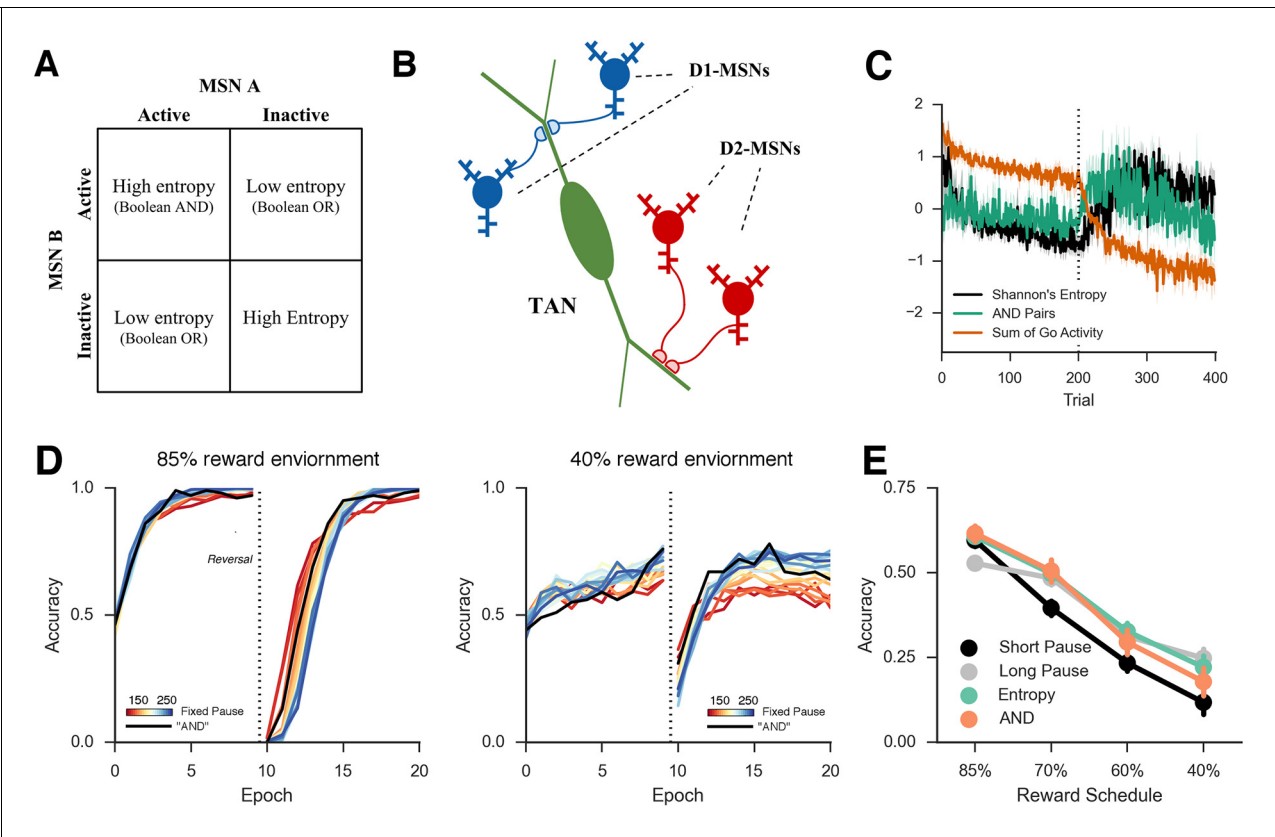

**Figure 9.** Approximations of MSN population uncertainty. (**A**) Shannon's entropy for two MSNs can be expressed with Boolean logic. Low entropy occurs when only one MSN is active (bottom left or top right box) and is the exclusive-OR function. High entropy is the logical opposite. (**B**) Spatially organized synapses could allow the detection of activity in two MSNs associated with separate motor responses, indicating high entropy. Activity in pairs of D1-MSNs signal high uncertainty given evidence for multiple responses, activity in pairs of D2-MSNs signal high uncertainty given evidence against multiple responses. (**C**) Detection of 'AND' pairs in Go population approximates Shannon's entropy across time, whereas simple summation of all Go unit activity does not. Dotted line denotes reversal at mid-point in training. (**D**) Neural network with AND detection performs well in both an 85% reward (*left*) and 40% reward (*right*) environments as compared to networks with fixed TAN behavior. (**E**) Varying pause duration with Shannon's entropy (green line) or the detection of AND conjunctions (orange line) results in similar behavior. MSNs, medium spiny neurons; TANs, tonically active neurons.

the striatum could transmit the population uncertainty (e.g., via further interactions between MSNs and fast-spiking interneurons). Our exercise below thus presents but a single plausible mechanism.

We simulated the network varying the TAN pause with AND detection in the Go units (see Materials and methods). In a network with a fixed duration TAN pause (180 ms) and trained on an 80% reward schedule, AND detection in Go units followed a similar temporal pattern to Shannon's entropy (*Figure 9*, center left). Both AND detection and entropy decline as the network learns the task and rise following the reversal of the reward contingencies. Not all summary statistics of MSN activity follow this pattern: for example, the simple summation of all activity in the Go population gradually declines over training and consequently, is not suitable for an approximation of entropy (*Figure 9*, center right). Notably, modulating TAN pause duration with the AND detection can also mitigate the behavioral trade-off between flexibility and stability, similar to the patterns observed using entropy (*Figure 9*, bottom right).

## Simulated ablation of M1-receptors

In the neural network model, varying the duration of the TAN pause alters the degree to which MSNs are disinhibited during reward feedback. Mechanistically, this depends solely on M2 receptors, which are responsible for inhibition of the TANs in the model, and does not involve changes in the activity of M1 receptors (see Materials and methods). However, previous empirical studies have linked M1 receptors to learning after reversal. Both TAN ablations and muscarinic antagonists impair reversal learning but do not affect acquisition in deterministic tasks (*Ragozzino et al., 2002*; *Witten et al., 2010*), an effect that is specific to M1 receptors (*McCool et al., 2008*).

To investigate the effects of M1 receptors in the model, we simulated selective M1 receptor ablations (see Materials and methods). In our simulations, the simulated ablation of M1 receptors only modestly decreases performance during acquisition, resulting in similar asymptotic accuracy as a control model that does not contain TANs (*Figure 3*, left). Following reversal, the effects are much more pronounced as simulated M1 ablation results in severely degraded accuracy (*Figure 3*, right). These results are qualitatively consistent with empirical findings showing impairment in reversal learning following M1 antagonists. It is noteworthy that the impairment with simulated M1 ablations only occur as a consequence of the disinhibitory effects of the TAN pause that drive MSN activity to a lower entropic state: control networks simulated without TANs do not show the same degree of impairment (*Figure 3*, right). Without this decrease in entropy or the increase in NoGo excitability, the network is able to learn following the reversal. However, both effects combine to allow the network to adapt more flexibly, perseverating in the correct action when it is rewarded but more sensitive to negative feedback facilitating reorienting.

## Simulations of post-pause TAN rebound burst

A phasic increase or rebound burst in TAN activity above tonic firing rates is commonly observed immediately following the reward-related pause (*Aosaki et al., 2010*). The functional significance of this burst is an open question but one likely function of the post-pause burst is to facilitate synaptic plasticity through the release of dopamine during an important time window. Optogentic stimulation of TAN neurons leads to the release of dopamine through the activity of the nicotinic receptors on striatal dopamine terminals (*Cachope et al., 2012*; *Threlfell et al., 2012*). Dopamine release precipitated by post-pause TAN activity would result in the delivery of dopamine immediately following a period in which striatal spiny neurons were disinhibited. This timing has important plasticity consequences as dopamine release following glutamatergic input promotes spine enlargement (*Yagishita et al., 2014*). Consequently, we hypothesize the post-pause TAN burst promotes plasticity by releasing dopamine during a sensitive time window following increased spiny neuron activity, while concurrently suppressing potentially interfering activity with muscarinic inhibition. To investigate the consequences of this hypothesis, we modeled the effects of a phasic increase in TAN activity following the feedback pause in the reversal learning task in an 85% reward environment. We found higher post-pause TAN firing rates resulted in higher asymptotic accuracy following reversal as compared to lower firing rates *Figure 10*. Interestingly, this effect was selective to reversal: there was no additional effect of of post-pause TAN activity on learning speed, both prior to and following reversal, and no effects on asymptotic performance prior to reversal. Together, the effects of TAN

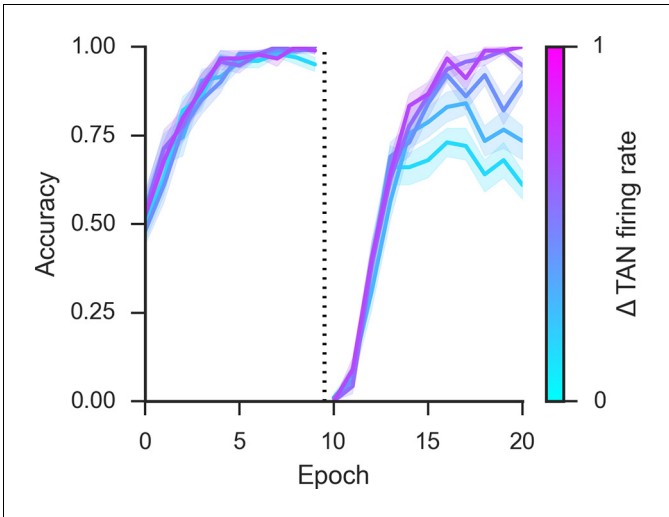

**Figure 10.** Post-pause TAN burst. An increase in phasic TAN activity following the feedback-related pause modulates asymptotic performance following reversal. Simulations shown with a fixed TAN pause of intermediate duration (190 ms) in an 85% reward environment, post-pause TAN firing rates are presented in normalized units of change relative over a baseline firing rate corresponding to the tonic firing rate. TAN, tonically active neuron.

pause and rebound burst act to enhance the BG network's ability to reverse and stabilize newly learned contingencies.

## Discussion

To act purposefully, a learner needs to ignore predictable irrelevant feedback driven by environmental stochasticity, while also flexibly adapting to a changing environment. This problem is general to learning with several proposed solutions (*Yu and Dayan, 2005*; *Behrens et al., 2007*; *Mathys et al., 2011*). Here, we proposed how the striatum may incorporate uncertainty to an adaptive process in line with these general principals of learning with a model in which striatal cholinergic signaling promotes encoding uncertainty in the MSN population. Across all simulations, incorporating TANs increased asymptotic performance over the previously published BG model (*Frank, 2005*). Simulated lesions of M1-AChRs revealed a qualitatively similar pattern of behavior as reported in rodent ablation and optogenetic studies (*McCool et al., 2008*; *Witten et al., 2010*). Parametric manipulations of TAN pauses revealed a trade-off between asymptotic accuracy and flexible learning driven by changes in MSN entropy. As a result, a network with any fixed TAN behavior can perform well in one task but may perform poorly in others. MSN entropy decreased over time in networks with long, fixed duration TAN pauses, making them less sensitive to stochastic noise, and facilitating performance in highly stochastic, sparsely rewarded environments. In contrast, networks with short, fixed duration TAN pauses had relatively constant levels of MSN entropy, resulting in labile representations sensitive to noise. As these networks are sensitive to negative feedback, they were able to learn quickly following a reversal, particularly in more deterministic environments.

The behavioral trade-off seen in the network is a product of a fixed learning behavior in a stochastic and non-stationary environment. Optimal behavior in such environments requires balancing the need to create precise estimates of an action's value with the flexibility to respond to sudden changes (*Yu and Dayan, 2005*; *Behrens et al., 2007*; *Nassar et al., 2010*). These goals are counterposed, thus any fixed learning rule that treats each observation equally at the time of observation will not be able to achieve both. If a learner is sensitive to recent events, she will be quick to detect changes but sensitive to any noise in the observations. Conversely, if a learner considers a longer time window to determine the best course of action, she will be slower to detect a change but be able to ignore irrelevant outcomes. Computationally, this pattern can be described by decay rate in the approximately Bayesian model as a fast decay rate considers recent events more strongly than a slow decay rate, which considers a longer history of events.

These findings suggest that TANs may provide an effective way of annealing the learning rate at the population level by implicitly coding uncertainty in the spiny neuron population code and making learning in the basal ganglia more (though not perfectly) Bayesian. Short TAN pauses are computationally similar to a high learning rate (fast decay in the Bayesian model) and long TAN pauses are similar to a low learning rate (slow decay). These effects are driven by relative differences in disinhibition of MSNs during reinforcement, and are not dependent on temporal dynamics of the pause (the same tradeoff between flexible and stable learning was seen when the magnitude of the pause rather than duration was manipulated). During the TAN pause, the reduction in TAN activity lowers the level of MSN inhibition, increasing the differential of activity between response selection and feedback. The greater disinhibition associated with longer TAN pauses results in a higher differential of activity and stronger reinforcement, driving the MSN population towards a lower entropy representation. Conceptually, embedding uncertainty into the population code of spiny neurons is similar to proposals that probabilistic population codes are used to maintain distributions for Bayesian inference (*Ma et al., 2006*). In both cases, the entropy of the population code can represent uncertainty, but here we explore the implication of this notion for learning within the basal ganglia, rather than inference in sensory cortex.

The reduction in MSN entropy with learning results in a lower population-level learning rate. This happens because the reduction in entropy reflects a reduction in the number of units active during reinforcement, such that the highly active units representing the dominant (most rewarded) action will persist in the face of spurious outcomes. Highly active units require more reinforcement to unlearn the association while inactive units are not available for reinforcement due to activity-dependent plasticity. As a result, an MSN population with lower entropy learns more slowly. As previously pointed out with the basal ganglia acetylcholine-based entropy (BABE) model, a reduction in MSN entropy will also decrease stochastic exploration as a decrease in MSN entropy lowers noise in the evidence for action (*Stocco, 2012*). In this work, we provide a single mechanism that affects learning rate and exploration as a function of entropy, but we note that even the exploration effect is learned and is collinear with the change in learning rate: we manipulated TAN pauses only during reinforcement, the window in which pauses overlap with phasic dopamine responses (*Morris et al., 2004*). Both changes arise as changes in synaptic weights alter the entropic representations in future trials, allowing the network to anneal its learning rate and change its stochastic exploration adaptively with its own uncertainty.

Given the size and extensive connectivity of TANs within the striatum, it is plausible that TANs are responsive to MSN entropy or a correlate, such as the coincident detection of pairs of active Go or NoGo neurons associated with separate motor responses. TANs receive substantial GABAergic innervation from MSNs (*Gonzales et al., 2013*) and are regulated by the neuropeptides enkephalin and substance P from MSN axons (*Gonzales and Smith, 2015*). This connectivity appears to be involved in learning as Pavlovian learning has been associated with an increase of δ-opioid receptor expression in TANs in rodent accumbens (*Bradfield et al., 2013*; *Laurent et al., 2014*). We have proposed, given this connectivity and the potential for MSNs to represent action selection uncertainty within a population code, it is plausible that a measure of uncertainty could be approximated directly with signaling from MSNs to TANs.

Differences in TAN pause duration have been reported in several cases that lead us to the hypothesis that the TAN pause duration may be an important modulator of cellular activity. In a behavioral context, differences in TAN pause duration have been reported to the motivational valence of a stimuli (*Ravel et al., 2003*) and to the presence or omission of reward (*Apicella et al., 2009*). In vitro, application of dopamine can elongate the TAN pause response (*Deng et al., 2007*) and the duration of the pause has been linked to the magnitude of initial excitatory input (*Doig et al., 2014*). These results suggest the duration of the TAN pause may be an important regulator mechanism and we further hypothesize that the local collaterals from many spiny neurons across the striatum onto a single giant TAN would allow the TAN to reciprocally modulate MSN population activity.

However, an important caveat to this hypothesis is that while the MSN to TAN synaptic connections are established and influence inhibitory currents within TANs (*Bolam et al., 1986*; *Chuhma et al., 2011*), we are not aware of direct in vivo evidence that TAN pauses are modulated by MSN activity. This constitutes a core prediction of the model. We propose that coincident MSN activity is largely involved in regulating the TAN pause that occurs during novel and/or rewarding

events (forming a window around phasic dopaminergic signaling) and may be less relevant at other periods of time. However, the exact influence of MSNs signaling on TAN activity and the mechanism for its transduction remain open to questions.

An alternate hypothesis is that pause duration is not modulated by MSN activity at all and reflects excitatory and dopamine input (*Ding et al., 2010*). In this case, we might still expect TAN activity to reflect uncertainty signals relevant to learning, relayed from other brain areas. The source of an uncertainty signal might originate through an explicit system that could signal a change-point in the reward rate (*Behrens et al., 2007*; *Nassar et al., 2010*; *Wilson et al., 2010*) or infer a change in a latent context or rule-structure (*Gershman et al., 2010*; *Donoso et al., 2014*). In principle, MSN entropy could also be influenced by hierarchical inputs from cortical areas that either further constrain action selection or increase noise, based on other factors such as perceived volatility. Human functional MRI (fMRI) studies have identified correlates of uncertainty processing in multiple areas, notably in the anterior cingulate cortex and striatum (*Behrens et al., 2007*; *Bach et al., 2011*), leaving open multiple possible mechanism to consider uncertainty at the whole brain level, including the top-down modulation of MSN entropy or TAN signals. Interestingly, a recent fMRI study have linked abstract prediction errors to the basal-forebrain, a separate source of cholinergic signaling (*Iglesias et al., 2013*). A final possibility is that uncertainty can be communicated to the TANs via more complex local striatal networks involving both MSNs and GABAergic interneurons, or an external source, such as a derivative of the dopamine learning signal. Nevertheless, while we have proposed that MSN synapses on TANs may directly relay an uncertainty cue, the core model prediction that TANs promote adaptive striatal learning via some representation of uncertainty is not itself sensitive to the source of the uncertainty cue.

Accordingly, the model makes several novel empirical predictions. A central prediction of the model is that MSN entropy should correlate with decision uncertainty during a task, which interacts with task stochasticity, and should covary with the degree of TAN pause inhibition. Similarly, we predict TAN pause duration (or equivalently, TAN firing rate) should correlate with decision uncertainty. Whereas previous studies have found a link between TAN activity and reversal learning (*Ragozzino et al., 2002*; *Witten et al., 2010*), our model makes the more specific prediction that M2 blockade would correspond with a decrease in overall accuracy during the performance of a probabilistic reversal learning task with multiple reward schedules. Notably, these manipulations were performed in the medial striatum, an area typically thought to be more involved in reversal than acquisition (*Clarke et al., 2008*). It is an open question whether TAN ablations would affect acquisition performance in other striatal areas, and in fact our model does predict changes in acquisition depending on the probabilities of reinforcement. We also expect M2 blockade would affect the tradeoff between exploration and exploitation by interfering with the striatum's ability to anneal it's own exploration policy with uncertainty. We would expect this effect to extend to overtraining; M2 blockade may lessen the effects of overtraining on stochastic tasks. We would expect optogentic silencing of TANs entirely to have similar effects.

While there are several previously proposed solutions to the general learning problem posed here (*Yu and Dayan, 2005*; *Behrens et al., 2007*; *Mathys et al., 2011*), the model we provide here is not meant as an alternative to these high level descriptions. Rather, we propose how a specific system, the striatum, can modulate its own behavior adaptively in line with Bayesian accounts and provide an interpretation of the role of TANs, tying together electrophysiological, pharmacological, lesion and behavioral data. This poses an additional theoretical question, as there is evidence to suggest that people learn in an adaptive way when faced with change points (*Behrens et al., 2007*; *Nassar et al., 2010*), and in different contexts in a process that likely involves the frontal cortex (*Collins and Koechlin, 2012*; *Collins and Frank, 2013*): why would the striatum consider uncertainty in a heuristic fashion if other systems are able to consider uncertainty explicitly? One benefit such a heuristic system that approximates optimal behavior could offer would be a lack of complexity. Optimal adaptation in a changing environment requires knowledge of the generative process underlying change, a problem that is complex and potentially ill-posed. Inference over the generative model may be too costly in highly complex, rapidly change or novel environments. In such an environment, striatal uncertainty could still be available and alter behavior in an adaptive way, encouraging exploration and weighing recent feedback strongly.

## Limitations

While there is some evidence that the duration of the TAN response may be an important regulator of cellular activity, the duration of the TAN pause has often not been linked to signaled probability of reward (*Morris et al., 2004*). It is important to note that the uncertainty over which actions are selected is independent of signaled probability of reward. As such, we would not expect the TAN response to vary with signaled probability of reward in well-trained animals performing a task in which the reward contingencies do not change. If the animals have learned the task well, then the uncertainty about the frequency of reward in a fixed schedule will be low.

Furthermore, the predictions of our model are not sensitive to the choice of TAN pause duration as a parameter of interest. In the implementation proposed, the duration of the TAN pause controls the degree to which spiny neurons are disinhibited. The algorithm we have used to simulate learning in the neural network model is sensitive to differences in excitation between stimulus presentation and feedback and these changes in disinhibition result in changes to the degree weights are updated in the neural network model. The model would make the same prediction if we varied the overall firing rate during the pause or altered inhibition directly.

An additional limitation is the focus of the current work on the effects of TAN behavior on MSN excitability relative to other components of striatal cholinergic signaling. The initiation of the TAN response depends on both dopamine and thalamic signaling, (*Aosaki et al., 1994*; *Ding et al., 2010*) both of which convey behaviorally relevant information (*Montague et al., 1996*; *Matsumoto et al., 2001*) and which we have not considered in the current work. Rather, we have only considered the modulation of such signals by MSN activity via collaterals. TANs may play a role integrating thalamic and dopaminergic signaling, but we are unable to make predictions about changes in the TAN response motivated by cell signaling or receptor activation.

# Materials and methods

## Neural network model of the basal ganglia

The neural network model presented here was adapted from the basal ganglia model presented by *Frank (2006)* and implemented within the emergent neural network software (*Aisa et al., 2008*). The model is available on our online repository at http://ski.clps.brown.edu/BG_Projects/Emergent_7.0+/. The model uses point neurons with excitatory, inhibitory and leak conductance that contribute to an integrated membrane potential transformed into a rate-code. Learning in the model is accomplished with the Leabra algorithm (*O'Reilly and Munakata, 2000*) and a reinforcement learning version based on dopaminergic modulation of Hebbian plasticity (*Frank, 2005*).

The neural network model is an attractor network organized into layers of point neurons (units), in which layers represent neural structures (c.f. *Frank, 2006* for a more detailed description of the model). The network is dynamic across time, and units within the network have stochastic behavior. The membrane potential $V_m$ of each unit within a layer is updated at each cycle (discrete time-point) as a function of its net current $I_{net}$ and a time constant $\tau_{net}$:

$$\frac{dV_m}{dt} = \tau_m \times I_{net}$$

The net current of a unit is, in turn, a stochastic function of its excitatory, inhibitory, and leak conductances ($g$) updated at each cycle as follows:

$$I_{net} = g_e(t)\overline{g}_e(E_e - V_m) +$$
$$g_i(t)\overline{g}_i(E_i - V_m) +$$
$$\overline{g}_i(E_l - V_m) +$$
$$\dots$$

where $V_m$ is the membrane potential of the unit, $E_C$ is the equilibrium potential for current $c$, and where the subscripts $e$, $l$, and $i$ refer to the excitatory, leak, and inhibitory currents, respectively. The total conductance of each channel $g_c$ are decomposed into the constant and time varying components $\overline{g}_c$ and $g_c(t)$. Units in the network are connected via synaptic weights, which can be excitatory

or inhibitory. The total excitatory input/conductance $g_e(t)$ to a unit is a function of the mean of the product the firing rate of each sending unit $x_i$ and the corresponding synaptic weights $w_i$, a time constant $\tau_g$ and a scaling factor $\kappa$:

$$\frac{dg_e}{dt} = \tau_g \left( k \frac{1}{n} \sum_i x_i w_i - g_e(t) \right)$$

Inhibitory conductance is computed similarly, but applied to synaptic inputs that come from inhibitory neurons, while leak conductance does not vary with time. Gaussian noise ($\sigma$) is added to the membrane potential $V_m$ of a subset of units in the network (units in the motor cortex and substantia nigra layers, $\sigma =$. 0015 and. 002 respectively, sampled from at each cycle of processing). As the conductances of each unit vary across time and between trial to trial as a function of input activity, the rate-coded activity of each unit with a layer varies with its inputs and noise. This added noise induces stochasticity in network choices (and their latencies; Ratcliff and Frank 2012) for any given set of synaptic weights, encouraging exploration: noise in motor cortex induces changes in the degree to which a candidate response is active and hence subject to disinhibition by gating via striatum, and noise in the SNc facilitates within-trial variation in the balance between Go and NoGo unit activity, differentially emphasizing learned positive versus negative outcomes and their effect on gating. Together, these sources of variance yield stochastic choice and dynamics within the striatum shown in *Figure 2*.

The activity communicated to other units in the network, $y_j$, is a threshold sigmoid function of the membrane potential:

$$y_j(t) = \frac{1}{1 + \frac{1}{\gamma[V_m(t) - \Theta]_+}}$$

where $\gamma$ is a gain parameter and where $[X]_+$ is a function that returns 0 if $X \leq 0$ and $X$ otherwise. This function is discontinuous at $V_m(t) = \Theta$, and is smoothed with a Gaussian noise kernel ($\mu = 0$, $\sigma = 0.005$) to produce a softer threshold and represent the intrinsic processing noise in neurons:

$$y_j^*(x) = \int\limits_{-\infty}^{\infty} \frac{1}{\sqrt{2\pi}\sigma} e^{-\frac{z^2}{2\sigma^2}} y_j(z - x) dz$$

where $x$ is the value $[V_m(t) - \Theta]_+$ and $y_j^*(x)$ is the noise-convoluted activation.

The layers in the network represent neural structures within the basal ganglia and thalamus. As a first approximation, action selection in the neural network is mediated by two simulated populations of MSNs. The populations of 'Go' and 'NoGo' units, representing striatonigral ('D1') and striatopallidal ('D2') MSNs, respectively, receive excitatory input from a cortical layer corresponding to a unique stimulus. Both the Go and NoGo layer contain 18 units, half of which, through their downstream targets, are connected to one of two motor responses. Reciprocal connections with a layer of inhibitory interneurons (simulating the GABergic fast-spiking interneurons in the striatum) control the overall excitability of the two populations. Activity in Go units inhibits a population in the globus pallidus interal segment (GPi), which results in disinhibition of the thalamus. Activity in the NoGo units inhibits a population in the globus pallidus external segment (GPe), which results in disinhibition of the GPi and inhibition of the thalamus. If there is sufficient activity in the thalamus, it provides a bolus of activity to a corresponding motor cortical column, which can then inhibit its competitors via lateral inhibition, and an action will be selected. This process is stochastic at a network level and depends both on interactions between units that vary with time (learning) as well as noise in the network within trials. However, while this process has trial to trial variability, activity in the Go pathway will facilitate the selection of an action through its downstream effects on the thalamus while activity in the NoGo pathway will suppress response selection.

Learning in the model occurs through weight updating in corticostriatal synapses without a supervised learning signal. A combination of a Hebbian learning rule and a contrastive Hebbian learning rule are used to determine the weight updates. Variants of contrastive Hebbian learning are consistent with large scale simulations of spike-time dependent plasticity (*O'Reilly et al., 2015*) and serve as a simplifying assumption to a more detailed mechanism of synaptic plasticity. The Hebbian

component of the learning rule assumes that the coactivation of MSNs and their cortical inputs proportionally determines the synaptic weight change. The contrastive Hebbian component is determined by the difference in coactivation of pre- and postsynaptic activity across response selection ('minus phase') and outcome feedback ('plus phase'). The equation for the Hebbian weight change $\Delta_{hebb}w_{ij}$ between sending unit $x_i$ and receiving unit $y_j$ is defined:

$$\Delta_{hebb}w_{ij} = y_j^+(x_i^+ - w_{ij})$$

where $y_j^+$ refers to the activity of the receiving unit during outcome feedback ('plus' phase) and $x_i^+$ is the activity of the sending unit. The contrastive Hebbian weight change is

$$\Delta_{CHL}w_{ij} = (y_j^+ x_i^+) - (y_j^- x_i^-)$$

Where $y_j^-$ and $x_i^-$ are the activity of the receiving and sending units during action selection ('minus' phase). The contrastive Hebbian term is subject to a soft-weight bound to keep between 0 and 1:

$$\Delta_{sbCHL}w_{ij} = [\Delta_{CHL}]_+(1 - w_{ij}) + [\Delta_{CHL}]_- w_{ij}$$

where $[X]_+$ is a function that returns $X$ if $X > 0$ and 0 otherwise and where $[X]_-$ is a function that returns $X$ if $X < 0$ and 0 otherwise. The Hebbian and contrastive Hebbian terms are combined additively with a normalized mixing constant $\kappa_{hebb}$

$$\Delta w_{ij} = \varepsilon[\kappa_{hebb}\Delta_{hebb} + (1 - \kappa_{hebb})\Delta_{sbCHL}]$$

Dopamine acts as a training signal in the model, providing a phasic increase during feedback for correct responses and a phasic pause for incorrect responses. Dopamine is simulated to act through D1 receptors in Go units, increasing excitability. In addition, D1 receptors were simulated to enhance contrast by increasing the striatal unit's activation gain and activation threshold. This has the effect of increasing the activity of highly active Go units and decreasing the activity of weakly active units, increasing the signal-to-noise ratio. D2 receptors were simulated in NoGo units such that an increase in dopamine decreases NoGo excitability. As a result, a phasic increase in dopamine during feedback has the effect of increasing Go activity relative to NoGo activity while a phasic decrease in dopamine will have the reverse effect. Altering the activity of Go and NoGo units in response to phasic changes in dopamine during feedback alters coritcostriatal weights through the contrastive Hebbian component of the learning rule. This facilitates error driven learning without providing the network a supervised learning signal.

## TAN behavior

TANs are endogenously active in the absence of synaptic activity (*Bennett and Wilson, 1999*) and were modeled as a separate endogenously active layer in which the leak channel equilibrium potential $E_l$ was typically higher than $V_m$. Unlike striatal Go and NoGo units, the activity of TANs were not dependent on synaptic signaling from other units in the network and TANs were simulated with little stochasticity. During stimulus presentation and action selection, TAN activity was held constant across all trials in all simulations (*Figure 2*, top left). Following action selection, a TAN burst–pause was simulated during outcome feedback. The burst was simulated at the onset of feedback to mirror the TAN burst by transiently increasing $V_m$ above the equilibrium potential. The subsequent pause was generated with an accommodation current, which allows the initial burst of TAN activity to create a subsequent hyperpolarization. This simulates the after-hyperpolarization that has been found to follow a depolarization in TANs via calcium-dependent potassium current (*Wilson and Goldberg, 2006*).

An accommodation current $I_a$ drives the membrane potential toward a low value which is added to $I_{net}$: $I_a = g_a(t)\overline{g}_a(E_a - V_m)$. A high accommodation current has the effect of hyperpolarizing the neuron as a function of how active it has been, simulating gated ion channels that accumulated with activity and driving the membrane potential to a low value. Consequently, the initial high firing rate at the onset of reinforcement causes the activity-dependent accommodation current to hyperpolarize the TAN units, silencing them for a length of time during feedback. The accommodation current is updated at each time step as a function of its time constant, $\tau_a$:

$$\frac{dg_a}{dt} = \begin{cases} \tau_{g_a}\Big(1 - g_a(t)\Big) & if \ b_a(t) > \Theta_a \\ \tau_{g_a}\Big(0 - g_a(t)\Big) & if \ b_a(t) < \Theta_d \end{cases}$$

where $\tau_{g_a}$ is the time constant of the channel conductance, $\Theta_a$ and $\Theta_d$ are the activation and deactivation thresholds required to invoke accommodation, respectively. The basis variable $b_a$ is a time average of the activation state:

$$\frac{db_a}{dt} = \tau_{b_a}\Big(y_j - b_a(t)\Big)$$

where $\tau_{b_a}$ is the time basis variable time constant and $y_j$ is the unit activity. During periods of persistently high activity, the basis variable $b_a$ will increase in proportion to its time constant $\tau_{b_a}$. When the basis variable exceeds the activation threshold $\Theta_a$, the accommodation conductance will increase, resulting in a net decline in current and a lower TAN firing rate. This process simulates the $Ca^{2+}$ dependent potassium currents in TANs: the basis variable simulates the build up of $Ca^{2+}$ in the cell and the effects of the basis variable on accommodation conductance simulates the opening (or closure) of $Ca^{2+}$ dependent channels. During the subsequent pause, $\frac{db_a}{dt}$ is negative as activity is low, causing the accommodation current to subside as the basis variable falls below the deactivation threshold $\Theta_d$, resulting in an increase in $V_m$ at the end of feedback (post-pause rebound, *Figure 2*, bottom left).

The behavior of the TAN pause was manipulated by varying $\tau_{b_a}$. Networks with fixed TAN pauses were simulated by specifying a constant value of $\tau_{b_a}$ such that the duration of the TAN pause lasted a pre-specified duration. The duration of the pause is reported in milliseconds (ms), which converted from cycles (the base unit of time within the network) at an assumed rate of 10 ms/cycle (Ratcliff and Frank, 2012). Pause durations were simulated ranging from 120 ms to 280 ms in 10 ms steps. For networks with a variable TAN pause, we simulated the impact of inhibitory collaterals from MSNs onto TANs such that the time constant $\tau_{b_a}$ was proportional to MSN activity (using either entropy or a more realistic approximation thereof, see below). This embodies our assumption that MSNs do not directly induce the TAN pause (which is driven by external inputs, e.g. from thalamus) but modulate its duration via accommodation (potentially via calcium-dependent potassium currents; *Wilson and Goldberg, 2006*) leading to an MSN activity-dependent reduction in TAN activity via direct signaling.

For the initial simulations of an adaptive mechanism to control pause duration, $\tau_{b_a}$ was updated as a function of the Shannon's entropy across the population of Go units on a trial to trial basis. On each time step during stimulus presentation, the rate-coded activity of each Go unit was normalized such that the normalized activity of all Go units summed to one. The normalized activity was then treated as a decision variable for the purposes of calculating entropy: the sum of activity over all units that contributed to a single response was treated as the probability of the selection of that response:

$$p_a(t) = \sum_{i=1}^{n_a} y_i^a(t)$$

where $y_i^a(t)$ is the activity of unit *i* corresponding to action *a* at time *t*. This assumption conforms with other interpretations of population activity within typical network models, where a probability distribution can be created by normalizing activation levels within a finite set of units (*Hinton and Sejnowski, 1983*; *Rao, 2005*; *D'Angiulli et al., 2013*). While the model presented here is more biologically complex, the same principle applies when treating the firing rates of discrete units as a probability distribution over actions. Shannon's entropy was calculated with $p_a(t)$ and summing across all cycles within the action selection phase:

$$H = -\sum_{t}\sum_{a} p_a(t)\log_2 p_a(t)$$

High entropy indicates that there is more competition between actions in the Go units and low entropy indicates low competition. We leverage this same quantity for controlling TAN pauses via

adaptive feedback (as described above) and also as a statistic to approximate the uncertainty of the network over its action selection policy, and how this evolves across learning.

A second more biophysical feedback mechanism was simulated, where instead of using an analytical expression for MSN entropy, TANs directly approximate MSN population entropy through synaptic integration. In these simulations, co-activation of pairs of MSNs co-localized on TAN dendrites were assumed to drive changes in TAN pause behavior. Thus the TAN time-constant $\tau_{b_a}$ was adjusted as a function of supra-threshold activity in pairs of Go units, in which each member of the pair corresponded to a different action (*Figure 9*). The 9 Go units corresponding to each of the 2 available actions were organized into 9 pairs of units. The activity in each unit was threshold and if the activity in both units were above threshold, the pair was counted as having an 'AND conjunction' (as the detection of supra-threshold activity is equivalent to the Boolean function AND):

$$c_j = \begin{cases} 1 & \text{if } (y_j^k > \theta) \text{ AND } (y_j^l > \theta) \\ 0 & \text{otherwise} \end{cases}$$

where $c_j$ is the Boolean value (0 or 1) of the 'AND conjunction' for the pair of units *j*, $y_j^k$ and $y_j^l$ are the activities for the units associated with actions *k* and *l*, respectively, $\theta$ is the threshold value. The value of $c_j$ across over all the actions and across all cycles in the action selection phase of a trial:

$$H_{conj} = \sum_t \sum_j c_j(t)$$

The time constant $\tau_{b_a}$ was set as a function of $H_{conj}$ which accumulated during response selection and hence affected the duration of subsequent accommodation hyperpolarization during the pause induced by the initial burst in the subsequent reinforcement phase.

## Synaptic effects of TAN activity

TANs were simulated to modulate Go and NoGo activity largely through the activity of M2 and M1 muscarinic receptors. The effects of M1 and M2 receptors are relatively well understood (*Goldberg and Reynolds, 2011*) and thus suitable for modeling. M1 and M2 receptors have opposing effects on MSN excitability: M1 activity increases dendritic excitability of indirect pathway MSNs through the post-synaptic closure of Kir2 K+ channels (*Shen et al., 2007*), while M2 receptors inhibit glutamate release in presynaptic terminals of both direct and indirect pathway MSNs (*Calabresi et al., 1998*; *Ding et al., 2010*). Cholinergic signaling can also modulate MSN excitability through increased GABAergic inhibition (*Witten et al., 2010*). Crucially, these effects act at different time scales: M1 activity is longer lasting and slower to initiate than the pre-synaptic effects of M2 receptors (*Shen et al., 2005*; *Ding et al., 2010*). Thus, the different temporal dynamics receptor activity may interact with the time-course of the TAN response: whereas the TAN burst activates M1 receptors and increases excitability in indirect pathway MSNs, the subsequent TAN pause reduces M2-mediated presynaptic inhibition and postsynaptic GABAergic inhibition (*Gerfen and Surmeier, 2011*).

M2-like muscarinic receptors are located on pre- and postsynaptic glutamatergic afferents of MSNs have the effect of inhibiting corticostriatal glutamate transmission (*Goldberg and Reynolds, 2011*). The effects of M2-like receptors were modeled with inhibitory synaptic connections from TANs to the Go and NoGo units. Excitatory synaptic connections to inhibitory interneurons were also used to simulate the nicotinic stimulation on GABAergic interneurons (*Witten et al., 2010*). These interneurons project to both Go and NoGo units and have an inhibitory effect. The effects of simulated M2 receptors and excitatory connections to the interneurons results in increased inhibition of both layers during action selection when TAN activity is constant (Figure 2, left) and disinhibition during the feedback pause (Figure 2, center).

M1 muscarinic receptors were simulated with a transitory increase in the leak channel Equilibrium potential $E_l$ for NoGo units during feedback. This was done to simulate the effects of M1 activity on inward rectifying potassium currents (*Goldberg and Reynolds, 2011*). Although M1-mRNA is found in both striatopallidal and striatonigral MSNs, the effect was simulated in the NoGo Layer only as increased sensitivity to glutamatergic signaling is selective to striatopallidal MSNs (*Shen et al., 2007*). In order to simulate M1 antagonists, the transitory increase in leak channel equilibrium

potential in NoGo units during the simulated TAN pause was removed. No effects of M1 receptors were simulated during the action selection phase.

Where noted, a rebound post–pause phasic increase in TAN activity was also modeled. A phasic increase in TAN activity is commonly reported following the feedback related TAN pause (*Aosaki et al., 2010*). Mediated by nicotinic acetylcholine receptors located on dopamine terminals in the striatum, the release of acetylcholine associated with a phasic increase in TAN activity evokes dopamine release (*Cachope et al., 2012*; *Threlfell et al., 2012*). The release of DA during the post–pause phasic TAN burst may modulate cortico-striatal plasticity in relation to activity during the feedback related pause DA release promotes spine growth specifically when stimulated after spiny neuron activity (*Yagishita et al., 2014*). The effects of the post-pause TAN burst were modeled by simulating the modulation of dopamine release by nicotinic receptors through the modulation of the dopamine membrane potential during the plus phase. The observed dynamic range of effects were normed as change TAN firing rate above baseline from 0 to 1.

## Bayesian model

An approximately Bayesian reinforcement learning model was used as a computational level description of the behavior of the neural network model. For i.i.d. Bernoulli trials in a stationary task (a reasonable approximation of the task), the exact posterior distribution of expected values is a beta distribution parameterized with $\alpha$ and $\beta$ (*Daw et al., 2005*)

$$Q(s,a) \sim Beta(\alpha, \beta)$$

The parameters $\alpha$ and $\beta$ can be updated online after each trial with the following rules:

$$\alpha_{(t+1)} = \alpha_{(t)} + r_{(t)}$$

$$\beta_{(t+1)} = \beta_{(t)} + 1 - r_{(t)}$$

where the reward on the current trial, $r_{(t)}$, is either 0 or 1. Because the task is non-stationary, a multiplicative parameter $\gamma$ ranging from 0 to 1 was used to decay the parameters $\alpha$ and $\beta$ during the update:

$$\alpha_{(t+1)} = \gamma(\alpha_{(t)} + r_{(t)})$$

$$\alpha_{(t+1)} = \gamma(\alpha_{(t)} + 1 - r_{(t)})$$

This multiplicative decay parameter increases the flexibility of the learner and has been used previously to model human learning (*Daw et al., 2005*; *Doll et al., 2009*, *2011*). In simulations, $\gamma$ was either held constant throughout training (analogous to fixed pause duration) or varied as a function of trial-to-trial changes in uncertainty with two free parameters, $\gamma_0$ and $\gamma_1$:

$$logit(\gamma) = \gamma_0 + \gamma_1 \cdot \Delta H$$

The free parameters $\gamma_0$ and $\gamma_1$ determine the baseline decay rate and sensitivity to trial-to-trial variations in uncertainty. Importantly, $\gamma_1$ was negative as performance is best when the model decays more quickly (and hence learns more from individual outcomes) during conditions of high uncertainty/volatility, analogous to shorter TAN pauses with more MSN uncertainty. Uncertainty was defined as the Shannon's entropy of the action selection probability, determined by integrating the expected values of both actions over the belief distributions:

$$H = -\sum_{i=1}^{n} p(a|s) \log_2 p(a|s)$$

The difference in entropy between trials was smoothed using a delta rule algorithm with the learning rate $\eta$ as a parameter. $\Delta H(a)$ was initialized at the 1 bit (the maximum possible entropy) and updated after each trial with the following rule:

$$\Delta H \leftarrow \Delta H + \eta \delta$$

$$\delta = \Delta H - [H_t - H_{t-1}]$$

Smoothing the difference in entropy between trials reduces the influence of trial-to-trial variance and allows for entropy for previous recent trials to influence the decay rate.

Action selection was accomplished through sampling Q-values. This allows for the consideration of uncertainty during the action selection. For each trial, a Q-value from the belief distribution of each available action was randomly sampled. The action with the highest Q-value was selected for the given trial.

## OpAL

An algorithmic model of the basal ganglia adapted from the OpAL model (*Collins and Frank, 2014*) was used to provide a mechanistic description of the feedback-control mechanism proposed in the neural network. OpAl is an actor–critic reinforcement learning model that mimics the opponent (D1/D2) actor system in the neural network. In OpAL, a single critic learns the value of a stimulus with its own learning rate ($\eta_c$). A prediction error ($\delta$), the difference between the observed reward ($r$) on a trial and the learned value of the critic on trial ($V_t$), is used to update the estimate on each trial.

$$\delta = r - V_t$$

$$V_{t+1} = V_t + \eta_c \delta$$

The prediction error on each trial is also used to update to actor values, a 'Go' ($G_t$) and 'NoGo' ($N_t$) value, for the chosen action conditional on the current stimulus with two separate learning rates:

$$G_{t+1} = G_t + \eta_G \delta$$

$$N_{t+1} = N_t + \eta_N \delta$$

where $\eta_G$ and $\eta_N$ are the learning rates for the $G$ and $N$ weights, respectively. These two actor values correspond to the Go and NoGo units in the neural network and model learned contributions of striatonigral and striatopallidal MSNs to action selection. Choices between actions were made using a softmax policy choice on the linear combination of actor weights:

$$p(a) \propto \exp\{\beta_G G_a - \beta_N N_a\}$$

Here, $\beta_G$ and $\beta_N$ control the degree to which each of the weights influence action and $p(a)$ is the probability action $a$ is selected. *Collins and Frank (2014)* simulate a variety of documented effects with this model, including how dopamine affects the asymmetry in learning and choice incentive (sensitivity to gains vs costs of alternative actions), show how its behavior converges to expected values and provide a normative interpretation.

Here, we expand OpAL to include the effect of TANs on modulating learning rate, using a multiplicative decay term, $\gamma$. On each trial, following the update of the actor weights, the weights were decayed to a naïve prior with the inverse logit transform of $\gamma$ as follows:

$$G_{t+1} \leftarrow G_{t+1}\left(\frac{1}{1+e^{-\gamma}}\right) + 0.5 \times \left(1 - \frac{1}{1+e^{-\gamma}}\right)$$

$$N_{t+1} \leftarrow N_{t+1}\left(\frac{1}{1+e^{-\gamma}}\right) + 0.5 \times \left(1 - \frac{1}{1+e^{-\gamma}}\right)$$

The inverse logit transform of $\gamma$ range bounds the decay rate between zero and one. Gamma was either held fixed, simulating fixed duration TAN pauses, or was allowed to vary with as a linear function of entropy in the model. Entropy in the model was the Shannon's entropy of the policy function:

$$H = -\sum_a p(a)\log_2 p(a)$$

## Simulated task

Both the neural network and Bayesian models were trained on a series of two alternative forced choice tasks. In each trial, the models were presented with one of two stimuli and made one of two responses. Equal valued rewards were delivered pseudo-randomly on a fixed stochastic schedule over epochs of 20 trials. The reward schedule for the two options were not yoked, such that sampling one action did not provide full information of the other's reward schedule. After 10 epochs of

20 trials (or 200 trials), the reward contingencies for each action were exchanged and the models were trained on an additional 10 epochs.

For both the neural network and the Bayesian models, each parameterization was trained on 50 instantiations. Initial weights of the neural network were randomized for each simulation. Simulations were run over eight reward schedules. The payout schedules for the best action were 85, 80, 75, 70, 65, 60, 55 and 40%. The payout schedules for the lowest rewarded action action were 15, 20, 25, 30, 35, 40, 45 and 10%. Comparisons between the neural network with and without TANs, as well as the manipulation of pause duration, were shown in networks trained on an 80/20% reward schedule.

## Acknowledgements

We thank Anne Collins and Matthew Nassar for helpful comments on an earlier version of this manuscript.

## Additional information

### Competing interests

MJF: Reviewing editor, *eLife.* The other author declares that no competing interests exist.

### Funding

| Funder | Grant reference number | Author |
|---|---|---|
| National Science Foundation | # 1460604 | Michael J Frank |
| National Institutes of Health | R01 MH080066-01 | Michael J Frank |

The funders had no role in study design, data collection and interpretation, or the decision to submit the work for publication.

### Author contributions

NTF, Conception and design, Acquisition of data, Analysis and interpretation of data, Drafting or revising the article; MJF, Conception and design, Analysis and interpretation of data, Drafting or revising the article

### Author ORCIDs

Michael J Frank, iD http://orcid.org/0000-0001-8451-0523

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
