## [Decision Letter]

[Editors’ note: a previous version of this study was rejected after peer review, but the authors submitted for reconsideration. The first decision letter after peer review is shown below.]

Thank you for choosing to send your work entitled "TANs adjust striatal learning as a function of population uncertainty: Computational models in stochastic environments" for consideration at *eLife*. Your full submission has been evaluated by Timothy Behrens (Senior editor) and three peer reviewers, one of whom is a member of our Board of Reviewing Editors, and the decision was reached after discussions between the reviewers. Based on our discussions and the individual reviews below, we regret to inform you that your work will not be considered further for publication in *eLife* in its current form.

The reviewers felt that the modelling and results sections were not explained with sufficient clarity for them to properly evaluate the manuscript. They felt that the question was important and that the findings would potentially be important. However, on top of technical questions, there were substantial concerns about the clarity and articulation of the work, which meant that the reviewers were not comfortable offering a clear recommendation.

The reviewers all agreed that there were some specific aspects of the paper presentation that needed to be addressed should the authors wish to resubmit the paper:

1) The model and previous work were poorly explained, figures incomplete and the explanations assumed familiarity with the authors' previously published work.

2) It would be desirable to have more detailed figures of activity and connectivity in the network, rather than high-level summaries.

3) The relationship to experiments should be much better defined. Specific model explanations and predictions of experimental results should be presented.

4) The use of the terms 'entropy' and 'uncertainty' was unclear and needs to be better defined and motivated.

In addition, each of the reviewers felt there were specific gaps in the completeness of the study that weakened its conclusions. These are detailed in the reviewer comments. At least some of these specific suggestions should be incorporated, should the authors wish to resubmit the paper.

The two technical points that raised most concern in reviewer discussions were the points that relating to the effect of rebound phase after the TAN pause (Reviewer 3 point 1), and the possible mechanisms for computation of uncertainty or entropy within the network.

Reviewer #1:

This study pulls together existing models of reinforcement learning and several streams of experimental results to develop an interesting model of learning in a changing environment.

The study develops a model of a striatal learning, adding a critical component by asking what would happen if variability in the MSN pool could be treated as a measure of uncertainty about a changing environment. Under such circumstances, the authors show how modulation by TANs may allow the network to improve its learning rate depending on uncertainty.

Major comments:

1) I had some difficulty understanding how the TAN input acted on the MSN population. This was presented in the text in words, and in the figures through secondary readouts such as accuracy and entropy. I would like to see the simulated activity patterns (e.g., spike rasters) reported for TANS and MSNs in the various conditions. Additionally, an input-output relationship of TAN activity, pauses, and MSN excitability would help to clarify the physiological assumptions of the model.

2) As per Methods, the time constant of the TANs "was transformed such that pause duration in the network varied linearly with entropy."

I was disappointed that the study fell short of modeling how the TANs would compute the entropy of the network. This is just plugged in to the firing patterns. Given that the model incorporates the MSN population, it would have been nice to see a mechanistic implementation of the entropy computation step. The authors don't clearly state a mechanism for this: the closest sees to be a line in the Discussion:

"A measure of uncertainty could be approximated by summing the activity over multiple sources, either directly from MSNs or indirectly through other interneurons. "

I'm not sure how just summing activity will give this measure. I feel that the authors must suggest a more detailed mechanism for this essential step. I would strongly prefer that they also model such a mechanism; it would complete a missing link in the model.

Reviewer #2:

This paper addresses the question of reinforcement learning in an uncertain environment. The theoretical premise is that common reinforcement learning algorithms are not sensitive to uncertainty, and would not operate well in such an environment. They propose that Cholinergic interneurons in striatum can sense this uncertainty and modulate circuit dynamics and plasticity to make the system operate better in an uncertain environment.

Major concerns:

The notion of uncertainty is not well defined here. I suppose they mean an uncertain environment, but this in itself is still not well defined. It could be an environment that is not stationary as in a reversal task, or an environment in which reward is uncertain, that is a reward is given just with a certain probability. This is just the beginning of the problem here, because the uncertainty of the environment becomes confounded with the response distribution of MSN neurons.

This is a complex rate based network, and its behavior is not at all clear. This seems like a deterministic system, so why does it have any entropy? Entropy seems like an important concept here because somehow entropy of the network represents the uncertainty of the environment. I think what they do is that they make a histogram of (deterministic) network activity and use that as a probability distribution (even though the N dim state space is unique) and use that for the "Entropy" calculation. Note though that the network has no uncertainty. The term entropy here seems to obfuscate rather than clarify, but I think what they mean is that in an uncertain environment the network response is more spatially variable than in a certain environment. This leaves many questions. First, what is the origin of spatial variability in this network? I don't know how many units there are in this network, and have no idea why different units do not respond identically. Second, if there is this mapping from uncertainty to network variability, how does this happen? This is an example of a general problem with this paper, they run complex simulations and treat the results as an experimentalist would they describe the results, but seldom explain why the network generates these results.

The model needs to be rewritten. For example the acronym TAN is used in the title. How should a reader, who is not exactly in the same field, know what TAN's are? The Abstract is very technical, and can really only be appreciated by a very small group of scientists. Additionally, some statements that are made are too strong. For example the sentence: "The basal ganglia support adaptive reinforcement learning in accordance with algorithms that adjust action values using a static learning rate." Is this a fact, or a postulate supported by some evidence? Additionally, the rest of the paper is also hard to understand. The methods relay on previous papers, and there is never a good heuristic explanation of the network behavior. Even the training task is badly described, has no figures, and is mostly in the Methods section.

There is hardly any physiological data that this model is trying to account for, if it exists it is not evident in the paper. So this is a complex network, accounting for behavior, but it can receive little validation at the physiological level. Do we need this much complexity given the current state of the experimental data?

Reviewer #3:

This is an interesting manuscript, which incorporates ChIs into a basal ganglia model. ChIs are of ever-growing interest because of their increasingly apparent involvement in key functions of the basal ganglia and dysfunction in disease. The timely manuscript by Franklin & Frank is a follow-up study by this group (Frank, 2005). The authors incorporate striatal ChIs into their previous computational model which successfully mimics the basal ganglia function in normal and Parkinsonism conditions. In this study, the authors tested the role of ChIs in acquisition and reversal learning, by simulating M1, M2 receptormediated currents on medium spiny neurons (MSNs) and nicotinic receptor-mediated currents on GABAergic interneurons. The results replicated the experimental observations. In addition, the authors claimed that MSN entropy and ChIs pause feedback mechanism might control the pause duration which reveals a tradeoff between asymptotic accuracy and flexible learning.

This is an interesting and timely model but I have some concerns.

1) The major concern is that the model ignored the rebound phase of the pause response. The lack of this component will significantly lower the power and impact of the model. Compared to the initial excitation, which is only observed in about half of TANs, the rebound of the pause response, which has similar duration as the pause, is much more consistent observed across TANs (Aosaki et al., 2010). Combined with the facts that dopamine signals in striatum might last longer than phasic activities in cell bodies, and that the model only includes fast responses in MSNs, i.e. excitatory, inhibitory and leaking current, will the rebound of the pause response rewrite the learning process?

2) The authors propose a feedback loop from MSNs to ChIs. Although activating MSNs can induce inhibitory currents in ChIs, the currents are thought to be weak (Chuhma et al., 2011). Also the MSNs have very low firing frequency (Berke et al., 2004). Will these facts influence the conclusion of feedback loop of MSNs and ChIs? The conclusion that inhibitory currents from MSNs will slow the firing rate of ChIs is natural, especially from a model's point of view, but is that reasonable when compared with experimental data? Any justification, like the inhibitory current intensity, or average firing rate of MSNs needs to be provided.

3) The model used Go and NoGo units activities to represent the "weight for specific actions" and "the evidence against particular responses" respectively. At the same time, the model claimed that Go neurons are striatonigral MSNs and NoGo are striatopallidal MSNs, and further used their electrophysiological data, e.g. M1 only act on striatopallidal MSNs. This is a common setup of computational models, however, the reversal learning process, e.g. left v.s. right turn (Ragozzino et al., 2002), lever press for food v.s. water (Bradfield et al., 2013), are task choices rather than task stops. Is it appropriate to use Shannon's entropy across the population of MSNs here?

4) It would be useful to have the duration of pause in time rather than 15-25 (Figure 3, Figure 5, Figure 6).

[Editors’ note: what now follows is the decision letter after the authors submitted for further consideration.]

Thank you for resubmitting your work entitled "A cholinergic feedback circuit to regulate striatal population uncertainty and optimize reinforcement learning" for further consideration at *eLife*. Your revised article has been favorably evaluated by Timothy Behrens (Senior editor) and three reviewers, one of whom is a member of our Board of Reviewing Editors. The manuscript has been improved but there are some remaining issues that need to be addressed before acceptance, as outlined below:

The reviewers all felt that the paper was significantly improved from the previous submission, but that there were important aspects that needed to be addressed.

1) The model lacked a rebound phase in the pause response. The authors should analyze whether this might alter learning processes.

2) The authors should address the possibility that pause duration in TANs may simply reflect excitatory and DA input.

3) The authors must clear up a continuing lack of clarity about how uncertainty maps to entropy, stochasticity, and to sparseness of the distributed response.

4) The authors should address the concern that the presented model is complex yet at the same time biophysically implausible. Can they embed the core concepts in a much simpler model?

5) The authors should make extensive improvements in clarity throughout the paper, as indicated in many comments by the reviewers.

6) The authors should also suggest some other simple testable predictions in addition to the ones mentioned in paragraph nine, Discussion.

Reviewer #1:

Significance:

This modeling study considers how striatal reinforcement learning may incorporate uncertainty about the environment into controlling learning rate. This allows the system to ignore chance fluctuations in reliable environments, but also to learn more rapidly in cases where the environment is changeable. The revision is considerably improved and addresses all the main concerns I had previously raised.

Major comments:

1) Figure 8 results are good to see, they take the argument full circle by showing that a purely network-based implementation is able to achieve high performance in changing environments by appropriate control of TAN behavior.

2) The implication of Figure 6 is a bit confusing, especially the last panel. How do the network weight differences map to the performance in Figure 5? What would be a good outcome from the point of view of performance? Is the black line close to optimal?

*Reviewer #2:*

This manuscript incorporates cholinergic interneurons (Chls) into a basal ganglia model built by the same group (Frank, 2005). It is good to see that authors incorporate striatal ChIs into a computational model and simulated the interaction between spiny projection neurons and TANs. With this new model, the authors successfully tested their hypotheses of 1) changing of the duration of 'pause' in TANs could affect the balance of asymptotic accuracy and flexible learning in striatum, and 2) the spiny projection neurons and TANs system showed self-tuning property in learning rate and optimizing performance across stochastic environment.

This is an exciting model with some concerns.

1) The missing of rebound phase of the pause response will significantly lower the power of the model. The initial excitation of the pause response, which is included in the model, is only observed in about half of TANs. However, the rebound of the pause response showed in all the reported pause response (Aosaki et al., 2010). While the dopamine signals last longer in striatum than the phasic activities in cell bodies, will the rebound component of the pause response re-write the learning process, which built only upon the initial excitation and pause, of the model?

2) The hypothesis of the duration of the pause in TANs affecting the learning process is clever. However, the original studies of the pause response only compare the durations of pauses that are induced by aversive and appetitive stimuli (Apicella et al., 2009; Ravel et al., 2003). As mentioned in the Discussion, the model would not be suitable for the animals that are well trained. It is a little bit hard to argue the interaction between spiny projection neuron and TANs play the important role in regulating pause duration. Would it be possible that the pause duration in TANs might only reflect the excitatory and dopamine input (Ding et al., 2010)?

Reviewer #3:

This version of the paper is improved compared to the previous one, but I still have some concerns.

The basic idea here is that learning should depend on the level of uncertainty. The assumption is that the level of uncertainty is represented by the network of MSN neurons. If they are sparsely activated this implied high certainty and if they are activated in a distributed manner this means uncertainty. A key phase is "treated as uncertainty". The equations for entropy are not for true entropy, and the stochasticity to the extent it exists plays little role here, they are simply quantifications of a sparse or distributed response.

One improvement here is that rather than modulating the TAN pause completely via an external and biophysically implausible mechanism the detection of the "uncertainty" is now more biophysical, however if I understood correctly the control of the TAN pause duration is still external and non-biophysical.

This is a very complex model, yet it is not biophysically plausible. I have the feelings that the general concepts could be embedded in a much simpler model that would not obscure the main points in a mountain of details, on the other hand if it were truly biophysical it could be more directly compared to experimental results. Therefore, this is neither a clear lucid theory, nor a reasonable computational model.

---

## [Author Response]

[Editors’ note: the author responses to the first round of peer review follow.]

*[…] The reviewers all agreed that there were some specific aspects of the paper presentation that needed to be addressed should the authors wish to resubmit the paper:*

*1) Model and previous work were poorly explained, figures incomplete and the explanations assumed familiarity with the authors' previously published work.*

*2) It would be desirable to have more detailed figures of activity and connectivity in the network, rather than high-level summaries.*

*3) The relationship to experiments should be much better defined. Specific model explanations and predictions of experimental results should be presented.*

*4) The use of the terms 'entropy' and 'uncertainty' was unclear and needs to be better defined and motivated.*

In addition, each of the reviewers felt there were specific gaps in the completeness of the study that weakened its conclusions. These are detailed in the reviewer comments. At least some of these specific suggestions should be incorporated, should the authors wish to resubmit the paper. The two technical points that raised most concern in reviewer discussions were the points that relating to the effect of rebound phase after the TAN pause (Reviewer 3 point 1), and the possible mechanisms for computation of uncertainty or entropy within the network.

We would like to thank the editor and reviewers for their encouragement as well as their critical comments and feedback. The review process was extremely useful for encouraging us to clarify what our model is positing. It motivated us to better articulate the key model features, but moreover to conduct several new analyses and simulations that reveal the underlying mechanisms, and which address the key issues highlighted in the first submission. We further demonstrate a plausible implementation of the feedback mechanism we had posited.

In more detail, we have made the following changes to the manuscript to address their concerns:

1) We added several new figures to better illustrate the assumptions of the model, the model’s behavior and its connectivity. These include: a schematic description of the effects of TAN signaling on MSNs (Figure 1), multiple individual examples of simulated activity for both MSNs and TANs (Figure 2), synaptic weight changes over time as a function of TAN pause duration and its regulation by our feedback mechanism (Figure 3) as well as changes in TAN behavior as a consequence of the proposed feedback mechanism (Figure 3).

2) We have clarified the assumptions and predictions of the model as they relate to experimental data. This includes a mechanistic account of how the model accounts for previous behavioral findings and a discussion of the proposed feedback mechanism in relation to experimental findings.

3) We have expanded our Discussion of uncertainty and entropy and more concretely defined both in terms of the model.

4) We have added new simulations of a biologically plausible mechanism for implementing the feedback mechanism, whereas our earlier model only assumed that TANs would have access to some analytical computation of entropy within the network (Figure 8). We argue these simulations demonstrate the feasibility of local consideration of uncertainty and show that such a mechanism is capable of adaptive behavior.

In addition, we have revised the manuscript for clarity and to address individual points raised by reviewers. We have also changed the title of the manuscript to “A cholinergic feedback circuit to regulate striatal population uncertainty and optimize reinforcement learning.” Below we include detailed point by point responses to reviewers’ comments.

*Reviewer #1:*

*1) I had some difficulty understanding how the TAN input acted on the MSN population. This was presented in the text in words, and in the figures through secondary readouts such as accuracy and entropy. I would like to see the simulated activity patterns (e.g., spike rasters) reported for TANS and MSNs in the various conditions on page 9 and 10. Additionally, an input-output relationship of TAN activity, pauses, and MSN excitability would help to clarify the physiological assumptions of the model.*

We have added a schematic depiction of the effects of TAN signaling on MSNs to Figure 1 and have added multiple individual examples of simulated activity for both TAN and MSNs (Figure 2). We show mean activity for individual simulations in lieu of spike raster plots as we feel these are more easily interpretable for rate-coded networks. In addition, we have clarified our language as to how the secondary readout of entropy is informative of the unit activity and added a visual comparison between a low entropy and high entropy population (Figure 2). More generally, we have been much more specific about the sense of entropy (across the population, as a marker of decision uncertainty) that we are using throughout the manuscript.

*2) As per Methods, the time constant of the TANs "was transformed such that pause duration in the network varied linearly with entropy."*

*I was disappointed that the study fell short of modeling how the TANs would compute the entropy of the network. This is just plugged in to the firing patterns. Given that the model incorporates the MSN population, it would have been nice to see a mechanistic implementation of the entropy computation step. The authors don't clearly state a mechanism for this: the closest sees to be a line in the Discussion:*

"A measure of uncertainty could be approximated by summing the activity over multiple sources, either directly from MSNs or indirectly through other interneurons. "

*I'm not sure how just summing activity will give this measure. I feel that the authors must suggest a more detailed mechanism for this essential step. I would strongly prefer that they also model such a mechanism; it would complete a missing link in the model.*

We agree with the reviewer that this is an important question and have included new simulations to address this critique (detailed in the new sub-section “A local mechanism for entropy modulation of TAN duration” in the results and summarized in Figure 8). We propose a mechanism by which pairs of MSNs synapse closely together on the dendrites of TANs in such a way that facilitates the detection of coincident MSN activity coding for alternative actions. This detection is the Boolean function AND and we argue that AND detection is biologically plausible, e.g. via non-linear dendritic summation, and can be used to approximate entropy. We implemented this mechanism in the neural network using AND detection to modulate the duration of the TAN pause and show that doing so results in very similar behavior as reliance on the analytical computation of Shannon’s entropy.

We note that this is just one plausible mechanism by which TANs could be sensitive to choice uncertainty across a population of MSNs, and we highlight other potential mechanisms in the text. The main important prediction of our model is that TAN pauses affect the striatum’s uncertainty and hence learning rate, and that if it also has access to a source of uncertainty (whatever the source) in modulating TAN pauses, it can adaptively respond to the tradeoff in flexibility vs stability.

*Reviewer #2:*

*[…] The notion of uncertainty is not well defined here. I suppose they mean an uncertain environment, but this in itself is still not well defined. It could be an environment that is not stationary as in a reversal task, or an environment in which reward is uncertain, that is a reward is given just with a certain probability. This is just the beginning of the problem here, because the uncertainty of the environment becomes confounded with the response distribution of MSN neurons.*

In our new manuscript, we have taken great care to clarify the sense in which the MSN population represents the uncertainty over its choice policy, how it does so, and the relation to activity in MSN subpopulations (Figure 2). In broad terms, we are concerned with the uncertainty about which action will be selected on an individual trial. We have defined this as the Shannon’s entropy over the probability mass function representing alternative actions (see text for elaboration).

In normative models, three sources of uncertainty are often considered in the decision making process: estimation uncertainty, expected uncertainty, and unexpected uncertainty (Payzan-LeNestour, et al 2013). Estimation uncertainty reflects the lack of knowledge about the outcome statistics; expected uncertainty reflects the predictable trial to trial outcome variability (risk), and unexpected uncertainty reflects a change in outcome statistics, i.e. when outcomes occur outside the predictive distribution. All three forms of uncertainty are meaningful, however, we have avoided the use of these terms because the sources of uncertainty interact during learning, and their definition, depend on the generative process.

The neural network does not track each source of uncertainty separately, but nevertheless represents the net uncertainty about which action to select. Such a measure decreases with accumulated knowledge (i.e., as estimation uncertainty decreases) but also increases when statistics change (unexpected uncertainty), and our feedback mechanism capitalizes on both of these components to adaptively modulate learning rate. Indeed, the trajectory of MSN entropy over learning and reversal is similar to the uncertainty (again quantified by Shannon’s entropy) within the Bayesian learner.

*This is a complex rate based network, and its behavior is not at all clear. This seems like a deterministic system, so why does it have any entropy? Entropy seems like an important concept here because somehow entropy of the network represents the uncertainty of the environment. I think what they do is that they make a histogram of (deterministic) network activity and use that as a probability distribution (even though the N dim state space is unique) and use that for the "Entropy" calculation. Note though that the network has no uncertainty. The term entropy here seems to obfuscate rather than clarify, but I think what they mean is that in an uncertain environment the network response is more spatially variable than in a certain environment. This leaves many questions. First, what is the origin of spatial variability in this network? I don't know how many units there are in this network, and have no idea why different units do not respond identically. Second, if there is this mapping from uncertainty to network variability, how does this happen? This is an example of a general problem with this paper, they run complex simulations and treat the results as an experimentalist would they describe the results, but seldom explain why the network generates these results.*

We appreciate the reviewer’s remarks as the description of the network and the mechanisms involved was not clear and we have substantially revised the manuscript. We have updated the text describing the function of the network and how it implements action selection by representing alternative action values in separate sub-populations, and their effects on downstream pathways. We clarified throughout the manuscript (and including the equation in the text where it is first introduced) that the entropy we refer to is that over action selection. Thus although the units in the model are rate-coded, the population nevertheless can be more or less ‘certain’ in that activity can be more or less restricted to just the most dominant action or can be distributed across multiple alternative actions. This would be true even for a purely deterministic network, where uncertainty could still evolve with learning (synaptic weight changes that then affect the relative distribution of activity). But we note here that the network is also not a deterministic system – there is stochastic noise added to membrane potentials of cortical and striatal units, which aid in producing stochastic choices (exploration) early in learning before synaptic weights have evolved. This also means that greater entropy is associated with greater choice stochasticity.

Perhaps even more importantly, we have also responded to this Reviewer’s comments to show more explicitly how the results manifest in terms of underlying mechanisms. We included examples of MSN and TAN population activity as a function of early and late learning and how they correspond to different levels of entropy (Figure 2). We have shown that the effect of TAN pauses on modulating MSN excitability during reinforcement is to alter the divergence in corticostriatal synaptic weights coding for the optimal vs. sub-optimal action, and that the TAN feedback-mechanism regulates this divergence (Figure 6) to stabilize performance after initial learning (as in networks with long pauses) but also remain flexible to change-points (as in networks with short pauses). These were very useful exercises and we feel the resulting manuscript is substantially clearer.

*The model needs to be rewritten. For example the acronym TAN is used in the title. How should a reader, who is not exactly in the same field, know what TAN's are? The Abstract is very technical, and can really only be appreciated by a very small group of scientists. Additionally, some statements that are made are too strong. For example the sentence: "The basal ganglia support adaptive reinforcement learning in accordance with algorithms that adjust action values using a static learning rate." Is this a fact, or a postulate supported by some evidence? Additionally, the rest of the paper is also hard to understand. The methods relay on previous papers, and there is never a good heuristic explanation of the network behavior. Even the training task is badly described, has no figures, and is mostly in the Methods section.*

We appreciate the reviewer’s comments and have made the appropriate changes for clarity throughout.

*There is hardly any physiological data that this model is trying to account for, if it exists it is not evident in the paper. So this is a complex network, accounting for behavior, but it can receive little validation at the physiological level. Do we need this much complexity given the current state of the experimental data?*

We agree that the system is complex and not fully understood. Nevertheless, there is substantial physiological data showing that distinct MSN populations code for action values in their firing rates (e.g., Samejima et al 2005) and that manipulating excitability of these Go and NoGo populations acts to bias animal choices accordingly (Tai et al, 2012). There is also physiological data showing that TAN pauses window the phasic dopaminergic signals during reinforcement (we now include a panel in Figure 1 of these data from Morris et al 2004 because they are central to the effects we simulate). There is further evidence that TAN signaling is relevant for modulating MSN synaptic plasticity and excitability (cites), and that ablation of TANs results in reversal learning deficits. Together with the vast literature linking basal ganglia function to normative reinforcement learning models, we believe it is appropriate to ask questions driven by physiology as well as by what we expect to find in adaptive behavior with the intent of making explicit predictions of physiology. Finally, our model makes further falsifiable, but as of yet untested, physiological predictions which we highlight in the Discussion.

*Reviewer #3:*

*1) The major concern is that the model ignored the rebound phase of the pause response. The lack of this component will significantly lower the power and impact of the model. Compared to the initial excitation, which is only observed in about half of TANs, the rebound of the pause response, which has similar duration as the pause, is much more consistent observed across TANs (Aosaki et al., 2010). Combined with the facts that dopamine signals in striatum might last longer than phasic activities in cell bodies, and that the model only includes fast responses in MSNs, i.e. excitatory, inhibitory and leaking current, will the rebound of the pause response rewrite the learning process?*

We agree that both the initial excitation and the rebound burst are interesting and relevant, in addition to the pause itself. We have modeled some of the effects of the initial excitation (via M1 signaling). However, as we understand it, the main concern highlighted here is that the rebound burst may offset the effects of the pause that we simulate. However, the key point is that TAN signaling modulates the effect of dopamine on the excitability of recently active MSNs. It has recently been shown that DA promotes spine growth associated with synaptic plasticity only in a narrow time window (0.3 to 2 seconds) following glutamatergic input to the MSN, and only for the specific spines that were excited (Yagashita et al., 2014, Science). Thus the effects of TAN pauses would be to further enhance the excitability of relevant MSNs in the precise window during which DA modulates plasticity. The subsequent TAN rebound burst could further enhance DA signaling (via nicotinic receptors) that occurs immediately after this increase in excitability, which would only further enhance the plasticity credited to the recently active MSN (even if that MSN is now inactive due to the burst – the key point is that it was active a few hundred milliseconds prior, as per Yagashita et al.).

*2) The authors propose a feedback loop from MSNs to ChIs. Although activating MSNs can induce inhibitory currents in ChIs, the currents are thought to be weak (Chuhma et al., 2011). Also the MSNs have very low firing frequency (Berke et al., 2004). Will these facts influence the conclusion of feedback loop of MSNs and ChIs? The conclusion that inhibitory currents from MSNs will slow the firing rate of ChIs is natural, especially from a model's point of view, but is that reasonable when compared with experimental data? Any justification, like the inhibitory current intensity, or average firing rate of MSNs needs to be provided.*

While Chuhma and colleagues found the effects of GABAergic signaling to be weak, these effects were found in vitro and were not measured during the TAN pause. We have proposed that modulation by MSNs may occur during the TAN pause, a period in which TANs may be more sensitive to inhibition due to a reduction in intrinsic pace making activity. Additionally, there may be an additional modulatory effect of TANs once the pause is elicited by other sources (for example, the thalamus) that simply extend the duration or its depth.

While MSNs have a low firing frequency, they are active in response to reward feedback (Asaad and Eskander, J Neuro 2011), the period of time when we have proposed TANs to be sensitive to MSN activity. If their activity is coincident with the TAN response, then the low firing frequency may not be as critical of an issue.

We would also note that this is but one potential mechanism and that uncertainty could also be conveyed to TANs through other mechanisms, such as cortical signaling as we highlight in the text. The main important prediction of our model is that TAN pauses affect the striatum’s uncertainty and hence learning rate, and that if it also has access to a source of uncertainty (whatever the source) in modulating TAN pauses, it can adaptively respond to the tradeoff in flexibility vs stability.

*3) The model used Go and NoGo units activities to represent the "weight for specific actions" and "the evidence against particular responses" respectively. At the same time, the model claimed that Go neurons are striatonigral MSNs and NoGo are striatopallidal MSNs, and further used their electrophysiological data, e.g. M1 only act on striatopallidal MSNs. This is a common setup of computational models, however, the reversal learning process, e.g. left v.s. right turn (Ragozzino et al., 2002), lever press for food v.s. water (Bradfield et al., 2013), are task choices rather than task stops. Is it appropriate to use Shannon's entropy across the population of MSNs here?*

This reflects a somewhat unfortunate historical labeling of the units in our model in terms of “Go” and “NoGo” because it may give the impression that the NoGo units are involved solely in stopping. But instead the NoGo units simply provide evidence against a response during a choice, allowing for the response with higher net difference in Go-NoGo activity to be selected, so that both populations are active during choice. Indeed, emprically both striatonigral and straitopallidal neurons are active during action selection (Cui et al., 2013 Nature), and activation of striatopallidal neurons acts to lower the action value of the specific response under consideration in favor of the alternative one, and not to directly inhibit the action irrespective of value (Tai et al., 2012, Nature Neuroscience).

Nonetheless, it was sufficient for our simulations to rely on Shannon’s entropy over only the Go units (i.e. only the positive evidence for each action). This measure of entropy is highest when there is evidence towards both choices but also analytically, entropy is high when there is no positive evidence for either response. Physiologically that would be conveyed by high activation of the NoGo units, which can be accommodated in the revised scheme (Figure 8) in which we consider AND conjunctions over both Go and NoGo units.

*4) It would be useful to have the duration of pause in time rather than 15-25 (Figure 3, Figure 5, Figure 6).*

We have converted our pause durations from cycles of membrane potential updating (the base unit of time in the network) to milliseconds.

[Editors' note: the author responses to the re-review follow.]

*The reviewers all felt that the paper was significantly improved from the previous submission, but that there were important aspects that needed to be addressed.*

*1) The model lacked a rebound phase in the pause response. The authors should analyze whether this might alter learning processes. 2) The authors should address the possibility that pause duration in TANs may simply reflect excitatory and DA input. 3) The authors must clear up a continuing lack of clarity about how uncertainty maps to entropy, stochasticity, and to sparseness of the distributed response. 4) The authors should address the concern that the presented model is complex yet at the same time biophysically implausible. Can they embed the core concepts in a much simpler model? 5) The authors should make extensive improvements in clarity throughout the paper, as indicated in many comments by the reviewers. 6) The authors should also suggest some other simple testable predictions in addition to the ones mentioned in paragraph nine, Discussion.*

We have made the following changes to address the reviewers’ concerns:

1) We have added simulations of a post-pause rebound phase in the section “Simulations of Post-Pause TAN Burst” of the manuscript, focusing on the effects to which we had previously only discussed conceptually – that TAN bursts reciprocally modulate DA release via nicotinic receptors on DA terminals. We show how this effect further enhances the network’s ability to reverse in stochastic environments by amplifying phasic DA signaling associated with newly correct actions; interestingly, the effects of this modulation were selective to reversal.

2) In our Discussion, we have addressed the possibility that TAN pause duration may reflect excitatory and DA input rather than (or in addition to) local modulation by MSNs, and note that this possibility does not invalidate many of the predictions of the model.

3) We have further clarified our definition of uncertainty and entropy in the neural model. We have also included additional detail of the basic assumptions of the model and the biophysical mechanism by which MSNs adjust TAN pause duration (via modulation of after-hyperpolarization) to provide additional clarity.

4) We have added new simulations to summarize the main contributions of the TANs, supplementing the implementational level neural model and the computational level Bayesian model with a previously described opponent actor learning model (OpAL) that provides an intermediate, algorithmic level of description. With this model, we were able to summarize much of the functional properties by which TANs modulate learning, embedding the core concepts in a simpler model, as suggested. These simulations are presented in Figure 8.

5) We have added additional empirical predictions of the model.

In addition, we have revised the manuscript for clarity and addressed the individual points raised by the reviewers.

Reviewer #1:

*1) Figure 8 results are good to see, they take the argument full circle by showing that a purely network-based implementation is able to achieve high performance in changing environments by appropriate control of TAN behavior.*

We thank the reviewer for their encouraging comments.

*2) The implication of Figure 6 is a bit confusing, especially the last panel. How do the network weight differences map to the performance in Figure 5? What would be a good outcome from the point of view of performance? Is the black line close to optimal?*

Thank you for allowing us to further clarify this analysis, which gives insight into the mechanism by which TAN pauses modulate learning rate in MSNs. The weight difference in Figure 6 shows how the Go weights for one action over the other diverge across time; greater weight divergence translates into more deterministic choice.

Thus the weight divergence prior to reversal relates to asymptotic accuracy but is inversely related to reversal speed as more evidence is required to counteract the initial divergence. The black line shows that the feedback mechanism, whereby TANs are modulated by network entropy, allows the network to adapt so that divergence does occur (facilitating asymptotic accuracy) but is tempered to prevent overlearning, and thus also facilitates rapid reversal. This is analogous to the decay of the hyperparameters in the Bayesian model. In response to Reviewer 3, we have further adapted a simpler intermediate algorithmic model of striatal learning which summarizes the learning dynamics of the neural network (Collins & Frank, 2014), which similarly captures these weight divergences and feedback modulation thereof. This model is presented briefly as a complement to the network implementation and is described in the section “Normative descriptions of TAN pause behavior”.

Reviewer #2:

*1) The missing of rebound phase of the pause response will significantly lower the power of the model. The initial excitation of the pause response, which is included in the model, is only observed in about half of TANs. However, the rebound of the pause response showed in all the reported pause response (Aosaki et al., 2010). While the dopamine signals last longer in striatum than the phasic activities in cell bodies, will the rebound component of the pause response re-write the learning process, which built only upon the initial excitation and pause, of the model?*

We thank the reviewer for their encouraging comments. We agree that the pause rebound likely plays an important role in the learning process, separately from the effects of the pause itself (which still comprises the main focus of the paper).

We have now included simulations examining the effects of the post pause rebound in the new section “Simulations of Post-Pause TAN Burst”. We focus on the effects to which we had previously only discussed conceptually – that TAN bursts modulate DA release via nicotinic receptors on DA terminals (Threlfell et al., 2012; Capoche et al., 2012) which facilitates spiny neuron plasticity as a consequence of glutamatergic synaptic activity in the immediately preceding time-window (Yagishita et al., 2014). We find that this effect further enhances the network’s ability to reverse in stochastic environments by amplifying phasic DA signaling associated with newly correct actions; interestingly the effects of this modulation were selective to reversal. This does not however change the qualitative pattern of results reported elsewhere in the paper, related to the TAN pauses and modulation of learning rate.

*2) The hypothesis of the duration of the pause in TANs affecting the learning process is clever. However, the original studies of the pause response only compare the durations of pauses that are induced by aversive and appetitive stimuli (Apicella et al., 2009; Ravel et al., 2003). As mentioned in the Discussion, the model would not be suitable for the animals that are well trained. It is a little bit hard to argue the interaction between spiny projection neuron and TANs play the important role in regulating pause duration. Would it be possible that the pause duration in TANs might only reflect the excitatory and dopamine input (Ding et al., 2010)?*

While we maintain that weak inhibitory effects of individual MSNs on TAN would conspire to have observable effects across the population, particular as entropy in multiple MSNs grows, we agree that it is possible that the TAN pause duration could only reflect the excitatory (e.g., thalamic) and dopaminergic input. In this case, the downstream effects of TANs on MSN excitability and modulation of learning rate could still hold, as long as these excitatory signals provided to TANs covaried with a measure of uncertainty. fMRI correlates of an uncertainty signal have been identified in the frontal cortex and anterior cingulate (Behrens et al., 2007; McGuire et al. 2014), leaving open an alternative possibility that cortical inputs to TANs signal uncertainty. Were this to be the case, our hypothesized mechanism of local MSN entropy feeding into TANs would be wrong, but the reciprocal modulation of MSN entropy and learning by TANs – and hence many of our core predictions – would remain. We mention these issues in the Discussion section.

*Reviewer #3:*

This version of the paper is improved compared to the previous one, but I still have some concerns.

*The basic idea here is that learning should depend on the level of uncertainty. The assumption is that the level of uncertainty is represented by the network of MSN neurons. If they are sparsely activated this implied high certainty and if they are activated in a distributed manner this means uncertainty. A key phase is "treated as uncertainty". The equations for entropy are not for true entropy, and the stochasticity to the extent it exists plays little role here, they are simply quantifications of a sparse or distributed response.*

We have amended our Methods to clarify the role stochasticity plays in network behavior, the interpretation of the striatal activity as a discrete distribution over the set of actions, and our precise definition of uncertainty. In addition, we have added simulations of an actor-critic model that mirrors the neural network to better illustrate the parallel. We address each of these points below.

We maintain that the network does reflect uncertainty, quantified by an estimate of the entropy in the action selection probability, which we make explicit. We agree with the reviewer that individual neurons do not represent entropy in their stochastic firing (but see below re: stochasticity), but nevertheless, even a deterministic network can have uncertainty in its weights evidence for multiple actions. As such, we believe it is appropriate to treat the normalized activity of units in the striatal layers as a discrete probability distribution over the set of actions available to the network. We also note that the interpretation of normalized unit activity as encoding probability is common in network models (Hinton & Sejnowski, 1983; Rao 2005; McClelland, 2013) and we now refer to these papers for the basic assumption. Finally, note that the activity of striatal units are closely analogous to actor weights in an algorithmic actor-critic model. In an actor-critic model, actor weights reflect the tendency of the model to select an action but these weights are not constrained by expected value and approximate the probability an action will be selected (via a sigmoidal choice function). The striatal units share this property. As such, we treat the normalized striatal unit activity as an approximation of the discrete probability distribution over the set of actions, not due to a lack of stochasticity in the network.

It is also important to that the network does have several sources of stochasticity. A noise term is added to the membrane potential of several units, and random initial weights in the network makes this noise have a greater impact than when multiple weights conspire to select the same response. Consequently, noise in the network can cause the network to settle in an attractor that results in a low valued response, even if the weights between the input and striatal layers are larger for units associated with the high-value response. The probability that the network will settle in a sub-optimal attractor will decline as the difference in the input-striatal weight becomes larger for the high value response relative to the low value response.

We have included the relevant section of the Methods below explaining the relationship between added stochastic noise to membrane potential and behavior:

“This added noise induces stochasticity in network choices (and their latencies; Ratcliff & Frank 2012) for any given set of synaptic weights, encouraging exploration: […] Together, these sources of variance yield stochastic choice and dynamics within the striatum shown in Figure 2.”

*One improvement here is that rather than modulating the TAN pause completely via an external and biophysically implausible mechanism the detection of the "uncertainty" is now more biophysical, however if I understood correctly the control of the TAN pause duration is still external and non-biophysical.*

We apologize for the lack of clarity. We have indeed modeled the control of the TAN pause duration biophysically through the modulation of an accommodation current in TANs by coincident activation of Go units (AND pairs). The accommodation current corresponds to the after-hyperpolarization following the initial TAN burst, and which is thought to contribute to the TAN pause (Wilson and Goldberg, 2006). Thus Go activity accumulates during response selection, modulating accommodation currents, such that their effects on TAN firing are observable once the feedback-related TAN burst triggers the hyperpolarization. We have amended our Methods to clarify how the accommodation current works in the network and its biological basis.

*This is a very complex model, yet it is not biophysically plausible. I have the feelings that the general concepts could be embedded in a much simpler model that would not obscure the main points in a mountain of details, on the other hand if it were truly biophysical it could be more directly compared to experimental results. Therefore, this is neither a clear lucid theory, nor a reasonable computational model.*

We take the reviewer’s concerns about model complexity and parsimony seriously. Although we maintain that the more complex network is useful – it represents a self-consistent theory about corticostriatal network function that also comprises many previously explored effects which have been repeatedly tested in the literature, including the roles of D1 and D2 neurons in learning and choice incentive, the role of the subthalamic nucleus in online modulation of decision threshold, etc, and now adds the role of the TANs within this network. We feel that cumulative science – building on prior work – is important for modeling. Nevertheless, we appreciate the point that some of the basic functions of the TANS could be captured in a simpler model which does not require all of these dynamics. Although we had already previously included the Bayesian interpretation of the network, which is much simpler, we now supplement this higher level normative description with an intermediate algorithmic RL model, building on our previously described opponent actor learning (OpAL) model which was designed specifically to summarize the functional properties of the neural network in learning and choice. We show that the OpAL model shows the same tradeoffs in learning and reversal as a function of how much the actor weights accumulate, with TANs acting to modulate the decay rate of these actor weights.

These simulations show that the TAN effects can be summarized with a single parameter, and that varying this parameter with model uncertainty produces the same performance benefit shown in the neural network, and is also paralleled by analogous changes in actor weights. We hope this simplified depiction will appeal to readers who prefer this level of description.

However, we still believe the complexity in the neural network is useful, over and beyond the above issues related to coherency of a theory of basal ganglia. For example, in the OpAL model, we are unable to directly simulate the well-understood properties of muscarinic receptors. We are able to summarize their effects, but only because we have observed these effects in the more complex neural network in which we are able to directly simulate biophysical properties. The network does make simplifying assumptions that leaves it more biophysically detailed than most striatal RL models but still less biophysically detailed than some other networks that focus on spike timing etc. The level of modeling afforded by our network allows us to ask questions about behavioral dynamics while still incorporating relevant biophysical detail. As such, the three models we have included in the paper addresses the role TANs play in feedback-based learning at different levels of analysis, with the Bayesian learner providing a computational description of the reversal learning problem, the neural network providing a biological implementation and the actor-critic model summarizes the algorithm implemented by the network.